# Mitofilin Heterozygote Mice Display an Increase in Myocardial Injury and Inflammation after Ischemia/Reperfusion

**DOI:** 10.3390/antiox12040921

**Published:** 2023-04-13

**Authors:** Yansheng Feng, Abdulhafiz Imam Aliagan, Nathalie Tombo, Jean C. Bopassa

**Affiliations:** Department of Cellular and Integrative Physiology, School of Medicine, The University of Texas Health Science Center at San Antonio, 7703 Floyd Curl Dr., San Antonio, TX 78229, USA

**Keywords:** Mitofilin/Mic60, mitochondrial DNA, reactive oxygen species (ROS), SLC25As solute carriers, mPTP opening, mitochondrial calcium retention capacity, cGas/STING/p-p65 pathway, inflammatory markers, ischemia/reperfusion injury

## Abstract

Mitochondrial inner membrane protein (Mitofilin/Mic60) is part of a big complex that constituent the mitochondrial inner membrane organizing system (MINOS), which plays a critical role in maintaining mitochondrial architecture and function. We recently showed that Mitofilin physically binds to Cyclophilin D, and disruption of this interaction promotes the opening of mitochondrial permeability transition pore (mPTP) and determines the extent of I/R injury. Here, we investigated whether Mitofilin knockout in the mouse enhances myocardial injury and inflammation after I/R injury. We found that full-body deletion (homozygote) of Mitofilin induces a lethal effect in the offspring and that a single allele expression of Mitofilin is sufficient to rescue the mouse phenotype in normal conditions. Using non-ischemic hearts from wild-type (WT) and Mitofilin^+/−^ (HET) mice, we report that the mitochondria structure and calcium retention capacity (CRC) required to induce the opening of mPTP were similar in both groups. However, the levels of mitochondrial dynamics proteins involved in both fusion/fission, including MFN2, DRP1, and OPA1, were slightly reduced in Mitofilin^+/−^ mice compared to WT. After I/R, the CRC and cardiac functional recovery were reduced while the mitochondria structure was more damaged, and myocardial infarct size was increased in Mitofilin^+/−^ mice compared to WT. Mitofilin^+/−^ mice exhibited an increase in the mtDNA release in the cytosol and ROS production, as well as dysregulated SLC25As (3, 5, 11, and 22) solute carrier function, compared to WT. In addition, Mitofilin^+/−^ mice displayed an increase in the transcript of pro-inflammatory markers, including IL-6, ICAM, and TNF-α. These results suggest that Mitofilin knockdown induces mitochondrial cristae damage that promotes dysregulation of SLC25As solute carriers, leading to an increase in ROS production and reduction in CRC after I/R. These effects are associated with an increase in the mtDNA release into the cytosol, where it activates signaling cascades leading to nuclear transcription of pro-inflammatory cytokines that aggravate I/R injury.

## 1. Introduction

Mitochondria play a critical role in cell death in response to ischemia/reperfusion (I/R) stimuli [1,2,3]. Mitochondria are well-known as major players in triggering cell death signaling cascade resulting in necrosis and/or apoptosis, which contributes to the pathogenesis of various diseases, including I/R injury. During ischemia, the immediate consequence of insufficient cellular oxygenation and nutrient metabolites is the inhibition of the electron transport chain that weakens both energy conservation and oxidative metabolism. In acute conditions, re-oxygenation of cells leads to the production of reactive oxygen species (ROS) and a profound alteration of mitochondrial Ca^2+^ homeostasis that triggers mitochondrial permeability transition (MPT), which is associated with the opening of mitochondrial permeability transition pore (mPTP) [4]. Mitochondria have a double membrane structure: the outer membrane (OMM) and the inner membrane (IMM). The IMM is folded inward to form layers and contains the respiratory chain machinery, which generates a membrane potential necessary for the production of ATP in mitochondria. The function of mitochondria is related to their structural integrity. Recently, mitochondrial cristae regulation has been related to numerous diseases, including but not limited to obesity, neurodegenerative diseases, osteoporosis, diabetes, osteogenesis, and cardiac dysfunctions [5,6,7]. 

Mitofilin is a ubiquitously expressed mitochondrial membrane protein [8] that was originally considered a heart muscle protein based on the high expression of its mRNA in rat hearts [9]. Mitofilin is a critical organizer of mitochondrial cristae morphology and, thus, indispensable for normal mitochondrial function [10,11]. Two cDNAs encode different isoforms of Mitofilin, and both isoforms are derived by alternative splicing and encode two proteins, *IMMT-1* and *IMMT-2,* of 88 and 90 kDa [12]. Mitofilin is part of a multi-subunit protein complex in the IMM, named mitochondrial inner membrane organizing system (MINOS or MICOS), that controls cristae morphology [11]. A study using double labeling indicate that Mitofilin co-localizes with mitochondria but not with Golgi or endoplasmic reticulum [8]. Mitofilin acts as a cristae controller to control the release of cytochrome C in response to apoptotic stimuli [13]. Using RNAi, Mitofilin knockdown induced fragmentation of the mitochondrial network and damage to the cristae [13]. In contrast, Mitofilin overexpression induces preservation of the mitochondrial structure, leading to the restoration of mitochondrial function and improvement of cardiac contractile function in the diabetic heart [6], as well as the promotion of cardiac hypertrophy [14,15]. However, until now, no Mitofilin knockout mouse model has been developed. Therefore, studies investigating the role of Mitofilin using Mitofilin-deleted mice have not been performed yet. We recently established a novel approach that elucidated Mitofilin’s crucial role in triggering mitochondrial permeability transition [16]. In fact, we reported that Mitofilin levels are reduced during reperfusion in a duration-dependent manner. Recently, we revealed that Mitofilin physically binds to Cyclophilin D (CypD), a regulator of mPTP opening, on a coiled-coil domain located in its C-terminus, and I/R stress disruption of that interaction favors mitochondrial dysfunction that initiates acute I/R injury. 

To determine the mechanism underlying Mitofilin-induced mitochondrial dysfunction, we performed mass spectrometry and uncovered that Mitofilin interacts with several IMM proteins, including ATPase α and β, Cyclophilin D, VDAC1, 2, 3, as well as several members of the mitochondrial carrier subfamily of solute genes (SLC25A 3, 5, 11, 22 and 42). SLC25A is a family of mitochondrial carrier proteins responsible for the import of various solutes across the IMM amongst all eukaryotes [17,18,19]. Among the SCL25A solute carriers, we identified that bind to Mitofilin, four of which play a role in the mechanism of the electron transfer chain (ETC) and ATP production. We studied whether Mitofilin depletion dysregulates these SLC25As solute carrier functions that might result in increased ROS production after I/R. Within cells, mitochondria are another source of DNA in addition to the nucleus. Human mitochondria DNA (mtDNA) is 16,569 bp in length and encodes 22 tRNAs, two ribosomal RNAs, and 13 polypeptides [20]. Damage or depletion of the mtDNA can affect the electron transfer chain, and ATP production, enhancing oxidative stress and inflammatory responses that favor tissue injury [21,22]. The displacement loop, the D-loop, is among the promoters capable of the initiation of the transcription of the heavy and light strands that are located in the main non-coding region of the mtDNA. Recent studies have determined the role of mtDNA release in several diseases, including acute kidney injury (AKI). However, the mechanisms responsible for mtDNA release after I/R are still to be clarified. Myocardial I/R injury triggers numerous pathological changes, including oxidative stress and inflammation [23]. mtDNA has been speculated to have a probable bacterial origin, which causes it to be sensed as “foreign,” suggesting that it is seen differently from “self” DNA in cells. Therefore, mtDNA release in the cytosol triggers a series of mechanisms that lead to augmented nuclear transcription of pro-inflammatory factors, including IL-6, TNF-α, and ICAM. Here, we studied whether the depletion of Mitofilin by the I/R stress triggers mechanisms that increase inflammation. Release of mtDNA into the cytosol and outside cells can activate different recognition receptors (sensors) and innate immune responses, including cGAS-STING, cytosol inflammasomes (AIM2 or NLRP3), and TLR9 [24,25,26]. Recent studies have found that activation of these three pathways increases inflammation [27,28,29]. After acute kidney injury, we recently reported that Mitofilin depletion results in the activation of the cGAS/STING/p-p65 pathway that favors the transcription of inflammatory cytokines in the nucleus. 

We investigated the impact of the downregulation of the Mitofilin gene in I/R injury using a mouse model. We report that after I/R injury, depletion of Mitofilin impairs cardiac functional recovery, increases myocardial infarct size, and promotes mitochondrial structural damage and dysfunction that reduces the CRC. We propose that the mechanism by which Mitofilin knockdown increases I/R injury involves dysregulation of SLC25As solute carriers that is related to increased ROS production and release of the mtDNA into the cytosol, where it activates the signaling pathways that promote the transcription of pro-inflammatory factors including IL-6, TNF-α, and ICAM in the nucleus, which subsequently, exacerbate the I/R injury. 

## 2. Materials and Methods

### 2.1. Experimental Protocols

All protocols followed the Guide for the Care and Use of Laboratory Animals (US Department of Health, NIH, Bethesda, MD, USA) and received UT Health Science Center at San Antonio Institutional Animal Care and Use Committee (IACUC) institutional approval. Protocols were conformed to the Guide for the Care and Use of Laboratory Animals: Eighth Edition (2011) from the National Research Council. Animals were housed in the animal-specific pathogen-free facility at UTHSCSA’s main campus in cages with standard wood bedding and space for five mice. The animals had free access to food and drinking water and a 12-h shift between light and darkness. The animals were selected randomly, and a blinded investigator performed the data analysis.

### 2.2. Animals

Age-matched male adult littermate wild type and full-body Mitofilin heterozygote mice (HET) that we created were used between 4–6 months of age. The mutants are a transgenic strain generated by pro-viral insertion of a recombinant retrovirus to interfere with RNA expression of the IMMT gene, as described in [12]. Animals were genotyped using tail DNA and the following primers (Table 2). 

### 2.3. Generation of IMMT Knockout Mice

Two gRNA targeting Immt intro 1/2 and intro 12/13 were identified utilizing the CRISPOR.Tefor.net guide RNA search tool. The sequences are 5′: AATGAGTACAAACATGCGAT and 3′: GTTGATGTCTAGTATACCCG. gRNAs (purchased from Synthego) were mixed with eSpCas9 nuclease (Sigma) and transduced into a 2-cell embryo (B6SJLF1) by electroporation. The 2-cell embryo was transferred into receipt female after overnight culture, and positive pups were then screened by a pair of PCR primers flanking exon 2 and exon 12. The deletion was confirmed by PCR amplification, producing a 334 bp amplicon, which confirmed the deletion between the two exons. 

### 2.4. Antibodies and Reagents

The materials used in all the studies were purchased from Sigma-Aldrich unless otherwise stated. Antibodies against the following targets were utilized in the different studies (Table 1): 

### 2.5. Langendorff Heart Perfusion

The protocol of isolated perfused mouse hearts was similar to that previously described in [30]. Briefly, mice were anesthetized using ketamine (80 mg/kg i.p.) and xylazine (8 mg/kg i.p.). Hearts were carefully removed from mice and arrested in the cold (4 °C) Krebs Henseleit bicarbonate buffer solution containing (in mM): glucose 11, NaCl 118, KCl 4.7, MgSO_4_ 1.2, KH2PO_4_ 1.2, NaHCO_3_ 25 and CaCl_2_ 3, pH 7.4. The hearts were retrograde-perfused with KH buffer bubbled with 95% O_2_/5% CO_2_ at 37 °C using the Langendorff apparatus at a constant rate (3 mL/min). After 30 min of equilibration, global ischemia was induced for 35 min by stopping buffer flow (maintained at 37 °C) followed by 30 min reperfusion (for mitochondrial studies and western blot analyses) or 120 min (for cardiac function and myocardial infarct size assessment). A heart was considered acceptable when it reached a minimum left ventricular developed pressure (LVDP) of 80 mmHg at the end of the basal perfusion (at the onset of ischemia). Sham hearts for myocardial infarct size and cardiac function assessments were not subjected to I/R but were perfused for 3 h, while sham hearts for mitochondria and Western blot analysis studies received 95 min of perfusion without an I/R insult.

### 2.6. Isolated Heart Functional Measurements

Cardiac function was recorded as previously described in our articles [31,32] using a 1.4F SPR-671 pressure-sensitive catheter (Millar, Inc. Houston, TX, USA) inserted into the left ventricle (LV) via a left atrial incision. Different parameters were recorded, including LV end-systolic pressure (LVSP), LV end-diastolic pressure (LVEDP), and heart rate (HR), were taken using Powerlab software (ADInstruments). The LV developed pressure (LVDP = LVSP − LVEDP), the Rate-Pressure Product (RPP = LVDP × HR), the maximum rate of rise of the LV pressure (dP/dt max), and the maximum isovolumetric rate of relaxation (−dP/dt min) were calculated from the heart function recordings at the end of reperfusion by a blinded investigator. 

### 2.7. Myocardial Infarct Size Measurements

At the end of 120 min reperfusion, hearts were harvested and cut into five transverse slices parallel to the atrioventricular groove. Heart sections were incubated for 10 min in 2% triphenyltetrazolium chloride at 37 °C to differentiate viable (red) from infarcted (white) heart tissue. Stained sections were then fixed with 4% paraformaldehyde before being imaged. Planimetry using Adobe Photoshop CS6 was used to quantify the necrotic area in the total LV by a blinded investigator.

### 2.8. Mitochondrial Isolation

Mitochondria isolation from mice hearts process was similar to that described previously in [33]. Sections were placed in isolation buffer A (in mM): sucrose 70, mannitol 210, EDTA 1, and Tris-HCl 50, pH 7.4, at 4 °C, and homogenized with the ratio of 0.1 g of tissue/mL of buffer A. The homogenate was centrifuged at 1300× *g* for 3 min in a Galaxy 20R centrifuge (VWR), and the supernatant was centrifuged again at 10,000× *g* for 10 min at 4 °C. The mitochondrial pellet obtained was then resuspended in isolation Buffer B (in mM): sucrose 150, KCl 50, KH_2_PO_4_ 2, succinic acid (used as substrate) 5, and Tris/HCl 20, pH 7.4. The total protein concentration of mitochondria was assessed using the BCA assay kit (Thermo Fisher, Waltham, MA, USA).

### 2.9. DNA Extraction and Quantification

The expression level of different proteins mentioned in Table 2 was assessed from the total extracted DNA. Total DNA was isolated from hearts using the DNeasy Blood and Tissue kit (Qiagen, Hilden, Germany), and concentration was measured using spectrophotometry (Biodrop, Holliston, MA, USA). mtDNA was quantified in relation to nuclear DNA by amplifying regions of the mitochondrial gene CoxII and the nuclear gene App1 as described in [34] using real-time quantitative PCR (qRT-PCR) and the following primers (Table 2). PCR samples along with PowerUp SYBR Green Master Mix (Thermo Fisher, Waltham, MA, USA) were run on a 7500 Fast Real-Time PCR system (Applied Biosystems, Waltham, MA, USA) to obtain ΔCT (cycling threshold values). 

### 2.10. Ca^2+^-Induced Mitochondrial Permeability Transition Pore (mPTP) Opening

The mitochondrial resistance to calcium-induced mPTP opening was measured using in vitro Ca^2+^ overload in isolated mitochondria assay as previously described [35,36]. Free Ca^2+^ concentration was recorded at the excitation and emission wavelengths set at 500 and 530 nm, respectively, using 0.1 µM of the Ca^2+^ sensitive dye, calcium green-5N (Thermo Fisher). Samples of isolated mitochondria (500 µg protein in 2 mL of Buffer B containing succinate as substrate) were incubated for 90 s in a fluorescence spectrophotometer (Hitachi) set at 35 °C with constant stirring. For experiments with cyclosporin A (CsA), mitochondria were incubated in 2 mL of Buffer B supplemented with 2 µM CsA. Pulses of CaCl_2_ (10 nmoles) were applied every 60 s to buffer B, which makes a fluorescence peak when bound reversibly to calcium green-5N dye, followed by a decrease as mitochondria take up the free Ca^2+^. As Ca^2+^ load increases, the extra-mitochondrial Ca^2+^ concentration starts to accumulate, a signal of lower capacity for mitochondrial Ca^2+^ uptake, eventually leading to the opening of mPTP and the massive release of Ca^2+^ from Mitochondria. The amount of Ca^2+^ required to trigger the mPTP opening was considered as the mitochondrial Ca^2+^ retention capacity (CRC) of the sample. The CRC assessment is, therefore, an indicator of mPTP sensitivity to Ca^2+^ [37]. CRC was expressed as nmoles of CaCl_2_ per mg of mitochondrial protein.

### 2.11. Mitochondrial ROS Measurement

Mitochondria ROS generation was assessed using a spectrofluorometer (Hitachi F2710, Schaumburg, IL, USA) at 560/590 nm (excitation/emission) in 100 µg of mitochondrial protein in 2 mL buffer C containing 20 mM Tris, 250 mM sucrose, 1 mM EGTA, 1 mM EDTA, and 0.15% bovine serum albumin adjusted to pH 7.4 at 30 °C with continuous stirring. Amplex red dye (1 μM) (Thermo Fisher) and horseradish peroxidase (0.345 U/mL) were used to monitor H_2_O_2_ production, an analog for ROS production. The level of H_2_O_2_ was calculated using a standard curve of the H_2_O_2_ concentration and fluorescence intensity. ROS production was measured after the activation of the ETC complex I using glutamate–malate (3 mM).

### 2.12. Western Blot Analysis

To determine the levels of proteins, equal concentrations of lysed tissue (whole heart lysate or the mitochondrial fractions) were loaded into 4–20% Tris-glycine gels (Bio-Rad, Hercules, CA, USA) as described in [38]. Electrophoresis was carried out for 90 min at 100 V of constant voltage, followed by blotting onto nitrocellulose membranes at constant 90 V for 80 min. Membranes were blocked with 5% blotting-grade blocker solution (BioRad), probed overnight with specific primary antibodies at 4 °C, and visualized using IRDye secondary antibodies and an Odyssey CLx digital imaging system (LI-COR Biotechnology, Lincoln, NE, USA).

### 2.13. Protein Identification/Relative Quantification by Mass Spectrometry

To identify mitochondrial proteins that interact with Mitofilin with mass spectrometry, mitochondrial proteins from normal wild-type mice were immune-precipitated using an anti-Mitofilin antibody. Proteins eluted from the Mitofilin pull-down were separated by SDS-PAGE prior to mass spectrometry analysis. Gels were run, and ~2 cm regions of interest in each lane were subdivided into slices that were individually reduced/alkylated and digested with trypsin. The obtained digests were analyzed by HPLC-electrospray ionization-tandem mass spectrometry on an Orbitrap Velos Pro (Thermo Scientific, Waltham, MA, USA). Mascot (Matrix Science, Boston, MA, USA) was used to search the UniProt_mouse database and a database of common contaminants. The resulting Mascot files for the gel slices in each lane were combined for subset searching of the identified proteins by X Tandem, cross-correlation with the Mascot results, and determination of protein and peptide identity probabilities by Scaffold (Proteome Software, Houston, TX, USA). Relative quantities were determined by spectral counting. The most important identified proteins are represented in the table.

### 2.14. Transmission Electron Microscopy

Mitochondrial morphology was analyzed in normal and I/R mice hearts immediately fixed in a phosphate-buffered solution of 4% formaldehyde with 1% glutaraldehyde and stored at 4 °C overnight as previously described in [39]. Heart tissue sections were washed with PBS, post-fixed in 2% (*wt*/*vol*) osmium tetroxide for 2 h at room temperature, and dehydrated in a graded alcohol series before being embedded in Eponate 12 medium. The blocks were cured at 60 °C for 48 h and 70 nm sections were cut using an ultramicrotome, mounted on Formvar-coated grids, and double-stained with uranyl acetate and lead citrate. Samples were analyzed and imaged using a JEOL 1230 transmission electron microscope. We classified mitochondria as ‘damaged’ when they had more than 50% disorganized/destroyed cristae structure. The mitochondrial area was calculated using ImageJ software. 

### 2.15. mtDNA Release in the Cytosol

The release of mtDNA in the cytosol was measured using the mtDNA isolation Kit from ABCAM (ab65321), which allows mtDNA isolation from cells and tissues in high yield and purity without contaminations from genomic DNA. Heart tissues were resuspended in the cytosol extraction buffer and homogenized in a dunce tissue grinder. The whole cell lysate was centrifuged for 5 min at 700× *g* to pellet nuclei and membranes. The supernatant was centrifuged at 12,000× *g* for 30 min. Mitochondria fractions were obtained from the pellet and lysed in mitochondrial lysis buffer for 10 min while the supernatant was considered as cytosol. An enzyme mix was added to both fractions according to the manufacturer’s recommendation and incubated for 60 min. Ethanol was thereafter added to samples and incubated for 10 min before final centrifugation for 10 min at 12,000× *g*. The retained pellet contained mitochondrial DNA. We used quantitative real-time PCR analysis of cytochrome c oxidase (Cox I and Cox II) genes to measure changes in mtDNA content, and β-actin was used as a control. Primers were obtained from Invitrogen (Carlsbad, CA, USA).

### 2.16. Statistical Analysis

Data presented in bar graphs are expressed as means, and error bars are the standard errors of the mean (±SEM) for a minimum of three independent experiments (*n* ≥ 3). Comparisons were conducted using the Student’s *t*-test and one-way ANOVA with post-hoc Dunnett’s or Tukey’s corrections for multiple comparisons, where appropriate, using Prism 8 (Graphpad Software, Boston, MA, USA). A difference of *p* < 0.05 was considered to be statistically significant.

## 3. Results

### 3.1. Mitofilin^−/−^ Mice Do Not Survive

In collaboration with the UTHSCSA mouse genomic facility at UTHSCSA, we created a Mitofilin/Mic60 heterozygote (Mitofilin^+/−^, HET) mouse, as shown in Figure 1A. The Genotype of Mitofilin^+/−^ mice was confirmed using Western blot analysis for functional Mitofilin^+/−^ mice protein (Figure 1B) in which the levels of Mitofilin was half (50%) in Mitofilin^+/−^ mice compared to littermates WT (100%) (Figure 1B). Analysis of the heart weight (HW) in Mitofilin^+/−^ mice showed that they exhibited similar heart weight (Figure 1C) and the HW/BW (BW: body weight) ratio (Figure 1D) compared to littermates WT. After eight generations of Mitofilin^+/−^ crossing, we only obtained Mitofilin^+/−^ or Mitofilin^+/+^ (WT) mice. These results indicate that full-body deletion (homozygote) of Mitofilin induces a lethal effect in the offspring (Figure 1E) and that a single allele expression of Mitofilin is sufficient to rescue the mouse phenotype in normal conditions.

### 3.2. Mitofilin^+/−^ Heart Mitochondria Exhibit Impaired Mitochondrial Dynamics in Normal Conditions

Mitochondria are organelles that are highly dynamic. They undergo coordinated cycles of fission and fusion, referred to as ‘mitochondrial dynamics,’ in order to maintain their architecture, including shape, distribution, and size. To determine whether Mitofilin downregulation affects mitochondria dynamics in normal conditions, we performed Western blot analysis in mitochondria fractions. We found that proteins involved in the regulation of both fission and fusion were decreased in Mitofilin^+/−^ compared to WT mice (Figure 2). In fact, the protein levels of MFN2 and OPA1 (mitochondrial fusion) and Drp1 (promotes mitochondrial fission) were all decreased in Mitofilin^+/−^ mice compared to WT mice. Note that Mitofilin downregulation did not significantly affect the level of Cyclophilin D and MIA40. These observations suggest that Mitofilin knockdown might affect mitochondrial dynamics. 

### 3.3. Mitofilin^+/−^ Hearts Display Reduced Functional Cardiac Recovery and Increased Myocardial Infarct Size after I/R

Using the WT mice hearts, we recently showed that Mitofilin levels are reduced during an early moment of reperfusion, and the levels of Mitofilin were inversely proportional to the extent of infarct size after I/R [16]. In order to determine the functional significance of Mitofilin downregulation in cardiac function in stress conditions, we perfused hearts from WT and Mitofilin^+/−^ mice using a Langendorff system and recorded cardiac function pre- and post-ischemia. The cardiac functional recordings were similar in both groups before the onset of the global normothermic ischemia (35 min); the hearts from Mitofilin^+/−^ exhibited reduced ability to recover from the insult (Figure 3A). Mitofilin^+/−^ mice hearts exhibited a much reduced cardiac functional recovery as measured with the rate-pressure product (RPP), dP/dt max, and −dP/dt min at the end of 120 min reperfusion relative to pre-ischemia. The RPP at the end of reperfusion in WT averaged 14,773 ± 432 mmHg/min, and it was reduced to 9456 ± 1323 mmHg/min in Mitofilin^+/−^ hearts (Figure 3B). Note that the RPP was similar in both non-ischemic groups (19,199 ± 815 and 19,410 ± 1309 mmHg/min in WT and Mitofilin^+/−^, respectively). The dP/dt max and −dP/dt min at the end of reperfusion in WT averaged 1653 ± 167 and −1325 ± 121 mmHg/s. It was reduced to 985 ± 35, and −920 ± 78 mmHg/s in Mitofilin^+/−^ hearts (Figure 3C,D). Note that the dP/dt max and −dP/dt min were similar in both non-ischemic groups (3232 ± 202 versus 3218 ± 314 mmHg/s and −2106 ± 134 versus −2300 ± 203 in WT and Mitofilin^+/−^, respectively). We further assessed myocardial infarct sizes in the hearts at the end of 2 h reperfusion. We found that Mitofilin^+/−^ mice hearts exhibited much larger infarcted regions, as shown by increased white area compared to WT hearts (53 ± 3 versus 82 ± 6% in WT) (Figure 3E,F). Myocardial infarct size was similar in both non-ischemic groups. Note that after I/R, the level of Mitofilin loss was more pronounced in HET mitochondria compared to littermates WT (Figure 3G). These results suggest a crucial role for Mitofilin in the deleterious mechanism of the cardiac response to I/R injury.

### 3.4. Mitofilin^+/−^ Heart Mitochondria Exhibit Damaged Cristae after I/R 

Mitofilin is a protein that resides in the IMM and controls the folding of cristae that are unique to mitochondria. To determine whether the increase in myocardial infarct size is associated with mitochondria damage, we used electron microscopy images of heart tissues. We assessed the damage of the mitochondrial cristae morphology in normal and after I/R conditions in both WT and Mitofilin^+/−^ mice. Mitochondria with fragmented or disrupted cristae with empty spaces (in the matrix) were considered damaged mitochondria. In non-ischemic hearts, we found that cristae morphology of mitochondria in Mitofilin^+/−^ mice was normal and similar to WT (4 ± 0.8% versus 3 ± 0.4%) (Figure 4A). However, following I/R, Mitochondria from Mitofilin^+/−^ mice hearts exhibited more damaged cristae morphology versus WT (87 ± 3% damaged mitochondria in Mitofilin^+/−^ versus 69 ± 4% damaged in WT; Figure 4B). This result suggests that Mitofilin might play an important role in stabilizing cristae structure in normal and stressful conditions. 

### 3.5. Mitofilin^+/−^ Mice Mitochondria Display a Reduced Calcium Retention Capacity Required to Induce mPTP Opening after I/R

Opening of the mPTP plays a key role in the mechanism of cell death by apoptosis and necrosis that can be triggered by either increased ROS production or calcium overload [40]. To assess the ability of mitochondria from WT and Mitofilin^+/−^ hearts to resist mPTP opening due to calcium overload in both normal and stress conditions, we assessed the calcium retention capacity (CRC) of isolated mitochondria. Tissues after I/R were obtained from the entire heart since, in the global ischemia model, all the myocardium is considered as an area at risk. As shown in Figure 4C, normal/non-ischemic Mitofilin^+/−^ mitochondria respond similarly to WT mitochondria in their ability to retain Ca^2+^ (300 ± 12 nmol/mg of mito protein in WT-normal versus 293 ± 7 nmol/mg of mito protein in Mitofilin^+/−^ normal), with a higher CRC reflecting greater resistance to mPTP opening. After I/R, CRC is expected to decrease as mitochondria will be suffering due to I/R stress and Ca^2+^ influx [39]. Consistently, we found that Mitofilin^+/−^ mitochondrial CRC was significantly reduced than WT CRC (Figure 4D). The post-I/R CRC values for Mitofilin^+/−^ averaged 137 ± 9 nmol/mg of mito protein while those for WT were 207 ± 13 nmol/mg of mito protein, hence a 32% decrease in WT but a 53% decrease in Mitofilin^+/−^ CRC. To further confirm the involvement of mPTP opening in the mechanism leading to increased myocardial infarct size after I/R in Mitofilin^+/−^, we treated WT and Mitofilin^+/−^ hearts with Cyclosporin A [41], well known to delay the opening of mPTP via its binding to Cyclophilin D, a regulator of mPTP opening [42,43]. Treatment with Cyclosporin A brought back mitochondrial CRC to respective sham levels suggesting that Mitofilin^+/−^ mitochondria display a higher sensitivity to calcium overload required to induce mPTP opening compared to WT mitochondria (Figure 4C,D). Together, these results suggest a crucial role for Mitofilin regulation in the mitochondria response to calcium overload after ischemic insult.

### 3.6. Mitochondria from Mitofilin^+/−^ Mice Are More Uncoupled and Produce More Reactive Oxygen Species (ROS) Following I/R

We found that following I/R, Mitochondria from Mitofilin^+/−^ mice hearts exhibit more damaged cristae morphology versus WT. We thereafter compared whether mitochondria from Mitofilin^+/−^ hearts were more uncoupled compared to those from WT in both normal and stress conditions. We assessed mitochondrial membrane potential (MMP) using the JC-1 dye in isolated mitochondria. As shown in Figure 5A, the ratios of green/red fluorescence intensity were similar in WT and Mitofilin^+/−^ mitochondria (100 ± 4% in WT-normal versus 88 ± 5% of mitochondria protein in Mitofilin^+/−^ normal). After I/R, MMP decreases as mitochondria suffer due to I/R stress. Consistently, we found that Mitofilin^+/−^ mitochondrial green/red ratio of JC-1 was significantly increased than WT MMP (Figure 5A). The post-I/R MMP values for Mitofilin^+/−^ averaged 605 ± 64% of mito protein while those for WT were 390 ± 63%, hence a 3.9 fold increase in WT but a 6.8 fold increase in Mitofilin^+/−^ MMP. Since an increase in the green/red ratio of JC-1 indicates a reduction in MMP, this result indicates that Mitofilin^+/−^ mitochondria are more uncoupled than WT mitochondria after I/R. To confirm that observation, we measured the ROS production on the complex I of mitochondria from WT and Mitofilin^+/−^ heart. We consistently found that both groups produced similar levels of ROS in normal conditions (non-ischemic) (118 ± 6 pmoles/min/mg of mitochondrial protein in WT sham versus 123 ± 21 pmoles/min/mg of mitochondrial protein in Mitofilin^+/−^ sham mitochondria). However, after the ischemic insult, Mitofilin^+/−^ heart mitochondria produced more ROS than WT mitochondria (152 ± 17 pmoles/min/mg of mitochondrial protein in WT mitochondria versus 253 ± 232 pmoles/min/mg of mitochondrial protein in Mitofilin^+/−^ mitochondria), (Figure 5B). This result shows a 1.3-fold increase in WT but a 2-fold increase in Mitofilin^+/−^ MMP. This result indicates that Mitofilin^+/−^ mitochondria are more uncoupled than WT mitochondria after I/R. ROS are generally considered to play a damaging role in the cell; however, they are also known to function as essential signaling molecules that can trigger cardioprotective mechanisms [44]. From our previous studies [31], we have reported an increase in ROS after stimulation of complex I in cardiac mitochondria subjected to I/R insult, causing damage in the myocardium. Together, these findings suggest that Mitofilin knockdown increases the production of mitochondrial pro-deleterious ROS after I/R.

### 3.7. Mitofilin^+/−^ Mice Display Increased Mitochondrial DNA Release and Inflammatory Factors after Kidney IR Injury

Mitochondrial stress causes damage and release of mitochondrial DNA (mtDNA) into the cytosol. Since Mitofilin^+/−^ mice display an increase in mitochondrial damage, we measured the level of cytosolic mtDNA after I/R injury to determine whether Mitofilin knockdown-induced increase in I/R injury mechanism involves the release of mtDNA into the cytosol, known to trigger inflammation. We found that the levels of mtDNA (represented by D-LOOP1) were similar in both sham groups. However, after the 35 min ischemia, the level of mtDNA release into the cytosol was increased in the WT heart in the function of reperfusion (15, 30, 60, 90 min) compared to the sham. However, we found that in Mitofilin^+/−^ mice, the level of mtDNA was significantly increased compared to WT-I/R respectively (Figure 6A). We thereafter studied whether the release of mtDNA into the cytosol triggers a sequential mechanism that leads to increased inflammation. Because Mitofilin knockdown increases mtDNA release, we determined whether this effect was associated with an increase in the production of pro-inflammatory markers. We found that the levels of pro-inflammatory factors, such as TNF-α, IL-6, and ICAM-1, were similar in both sham groups. However, after 35 min ischemia followed by 2 h, and 4h reperfusion, the levels of these inflammatory factors were increased compared to WT-sham mice. However, in Mitofilin^+/−^ mice, the levels of these inflammatory factors were further increased compared to WT-I/R (Figure 6B–D). Our findings indicate that Mitofilin knockdown in the mouse increases the release of mtDNA from mitochondria to the cytosol. We report that the rise in the release of mtDNA in the cytosol in the Mitofilin^+/−^ mice effect is associated with an increase in the nuclear transcription of pro-inflammatory makers after I/R injury.

### 3.8. Mitofilin^+/−^ Mice Display Dysregulated SLC25As Solute Carrier Function after I/R 

To determine the molecular mechanisms involved in Mitofilin knockdown-induced deleterious effect after I/R, we sought to identify the set of proteins that interact with Mitofilin in the IMM. Mass spectrometry on immunoprecipitated pull-downs of Mitofilin from WT cardiac mitochondria samples showed that Mitofilin interacts with a large number of mitochondria proteins that include ATP synthase subunits α and β, the mPTP regulator, Cyclophilin D (Ppif), the mitochondrial contact site and cristae organizing system (MICOS) subunits (19, 25, 26, and 27), SAMM50, VDAC1/2/3, glucose-regulated protein 75 (GRP75) [16], and interestingly several members of the mitochondrial carrier family (SLC25) including 3, 5, 11, 22 (Figure 7). However, most of the identified SLC25As solute carriers that are involved in the transport of molecules across the IMM were crucial for the respiratory chain process. We, therefore, compared the levels of these SLC25As in WT versus Mitofilin^+/−^ after I/R. As shown in Figure 8A–D, we found a dramatic dysregulation of these four SLC25As solutes carriers in Mitofilin^+/−^ mice compared to littermates WT. In fact, after 35 min ischemia followed by reperfusion for 30, 60, and 90 min, the levels of SLC25A3 subunit were progressively reduced, while the levels of SLC25A 5, 11, and 22 subunits were increased in Mitofilin^+/−^ mitochondria compared to respective WT group (Figure 7). Note that we have also identified other SLC25A subunits, including SLC25A12, 13, 20, and 42, that are not represented here. Together these results indicate that the mechanism of Mitofilin knockdown induced cardio-deleterious effects after I/R involve dysregulation of SLC25As solutes carriers that might result in increased ROS production.

### 3.9. Mitofilin Knockdown Does Not Significantly Affect the Transcription of Protein in Mitochondria

Mitochondria possess a 16 kb circular double-stranded genome that is responsible for the expression of 13 subunits of the electron transfer chain, the 22 tRNAs essential for their translation, and the two rRNAs required for the assembly of the mitochondrial ribosome [45,46]. We determined whether Mitofilin downregulation in the mouse could affect the mitochondrial transcription process. We found that the levels of the mRNA of most of the proteins transcript in mitochondria were not different in Mitofilin^+/−^ mice compared to littermates WT (Figure 9). However, the levels of ND4 and Tfam were upregulated in Mitofilin^+/−^ mice compared to littermates WT. This result indicates that Mitofilin downregulation does not significantly affect the mitochondrial transcription process.

## 4. Discussion

In this paper, we report that Mitofilin knockdown mice (Mitofilin^+/−^) exhibit reduced cardiac functional recovery, increased myocardial infarct size, and promoted mitochondrial structural damage and dysfunction after I/R compared to WT. The mechanism of this cardiac dysfunction is associated with an increase in mitochondrial sensitivity to calcium overload required to trigger the opening of mPTP. We postulate that loss in Mitofilin causes dysregulation of SLC25As solute carriers resulting in increased ROS production that promotes an enhancement of mtDNA release into the cytosol, where it activates a cascade of signaling that leads to nuclear transcription of pro-inflammatory markers, including IL-6, TNF-α, and ICAM, subsequently, exacerbating I/R injury. 

Mitochondria play a critical role in the mechanism of a wide range of cellular actions, which include but are not limited to energy production, calcium homeostasis, chemical signaling, and regulation of cell death [47,48]. Growing recognition points to the importance of mitochondrial proteins, especially from the IMM, in health and pathophysiology [3]. Mitofilin is one of these key IMM proteins, which is the core unit of the MINOS complex that is the critical organizer of mitochondrial cristae morphology and, thus, essential for normal mitochondrial function [11]. We recently found that Mitofilin knockdown in rat myoblasts by siRNA increases calpain activity that presumably leads to mitochondrial structural damage resulting in a critical decrease in mitochondrial function, which is responsible for the increase in apoptosis through an AIF-PARP pathway [49]. In isolated perfused hearts from WT mice, we revealed that the levels of Mitofilin were reduced after I/R injury during the early moment of reperfusion. We reported that Mitofilin reduction due to I/R insult disrupts the Mitofilin-CypD direct link in the IMM, resulting in an increase in mitochondrial sensitivity to Ca^2+^ overload, ROS production, and dissipation of mitochondrial membrane potential (MMP), which are responsible for mitochondrial dysfunction [16]. However, the role of Mitofilin knockdown in intact mice subjected to I/R injury still needs to be clarified. To this end, we created a full-body Mitofilin heterozygote mouse (Mitofilin^+/−^). We found that full body homozygotes deletion of Mitofilin induces a lethal effect in the offspring, as Mitofilin^−/−^ mice were not obtained after multiple crossings of Mitofilin^+/−^, suggesting a crucial role of Mitofilin in the survival of mice. We observed that mitochondrial dynamics is dysregulated in Mitofilin^+/−^ compared to WT mice in normal conditions; this effect is associated with slightly damaged mitochondria morphology in normal conditions. We found proteins involved in the regulation of both fission and fusion were decreased in Mitofilin^+/−^ compared to WT mice (Figure 2), suggesting that a decrease in mitochondria fusion in Mitofilin^+/−^ might be compensated by the reduction in mitochondria fission, which presumably allows mice to support the mitochondrial dynamics dysfunction. This observation is supported by the unchanged transcription of mitochondrial proteins (Figure 9) in normal conditions. However, after I/R, Mitofilin^+/−^ mice display an increase in myocardial infarct sizes (Figure 3A–D,G) and a reduction in the cardiac functional recovery (Figure 3E,F). These results support the idea that Mitofilin regulation might play a crucial role in the development of I/R injury. This observation supports our previous finding in WT mice, indicating that there is an inversely proportional relationship between the reduction in Mitofilin levels during reperfusion and the extent of myocardial infarct size caused by I/R insult [16]. The mechanisms responsible for the increase in Mitofilin loss after I/R are still not completely determined. We recently reported that the levels of Mitofilin at the end of ischemia are similar to sham (non-ischemic) [16], suggesting that factors activated during reperfusion might favor the catalytic activity of mitochondrial proteins. Another possibility is that ischemia might weaken the Mitofilin link to other components in the MICOS complex, which is perturbed and washed out during reperfusion. After a kidney I/R insult, we found an increase in RIP3 translocation into mitochondria from the cytosol, where it interacts with and favors Mitofilin degradation [50]. However, further studies are still needed to clarify all the mechanisms responsible for the increase in Mitofilin loss after I/R. 

Currently, the consensus point to a crucial role of Mitofilin in maintaining the cristae morphology of mitochondria. However, the mechanism by which Mitofilin downregulation (reduction) acts to increase I/R injury is not completely understood. Consistent with this statement, we observed that adult normal Mitofilin^+/−^ mice exhibit much similar mitochondrial morphology and cardiac function comparable to WT. However, following the I/R insult, these hearts have more damaged mitochondria (Figure 4), suggesting that Mitofilin’s impact is more prominent after I/R, as Mitofilin knockdown results in cardio-deleterious effects. I/R injury is strongly associated with the dysfunction of mitochondria *through* the dysregulation of the electron transfer chain, which collapses the ATP-generating membrane potential, an increase in ROS generation, and/or Ca2+ overload leading to the opening of mPTP [31]. Opening of mPTP plays a pivotal role in the mechanism of cell death after I/R injury [37]. Therefore, we defined the role of Mitofilin knockdown in regulating the opening of mPTP after I/R. The mitochondrial response to Ca^2+^ overload was measured using mitochondrial CRC. A higher CRC indicates healthier mitochondria that are more resistant to calcium-induced mPTP opening. Our results indicate that Mitofilin^+/−^ mitochondria from normal (non-ischemic) hearts exhibit similar CRC compared to WT mitochondria, but this CRC is considerably reduced after I/R in Mitofilin^+/−^ versus WT mitochondria(Figure 4C,D). This result suggests that during reperfusion, Mitofilin^+/−^ mitochondria are more susceptible to impairment in response to Ca^2+^ overload that promotes mPTP opening. When taken together, our results point to a critical role of the Ca^2+^ regulation in the mechanism responsible for mitochondrial damage in Mitofilin^+/−^ mice hearts after I/R.

The molecular composition of the mPTP is still not elucidated. However, the opening of mPTP results in a large pore that leads to mitochondrial depolarization and cell death [39]. Using several cell lines, a mutant TDP-43 mouse model, and human ALS-affected spinal cord samples, Maters’ group has reported that TDP-43 induces the release of mtDNA into the cytosol via mPT pore [51]. We, therefore, studied whether Mitofilin knockdown increases I/R injury by enhancing mtDNA damage and release in the cytosol. The release of mtDNA into the cytosol during I/R is known to act as pro-inflammatory signaling that mediates inflammation-related injury, thereby making mtDNA release in the cytosol a potent pro-inflammatory mediator [52]. In this study, we establish that Mitofilin+/− mitochondria release more mtDNA into the cytosol after I/R injury compared to littermate WTs (Figure 6A), which suggests that regulation of Mitofilin can affect the release of mtDNA into the cytosol after I/R. This finding indicates a potential role for mtDNA release into the cytosol in the cardio-deleterious mechanism of Mitofilin knockdown in I/R injury. Our results are supported by previous reports that found that inhibition of the release of mtDNA into the cytosol promotes EGCG-induced cardioprotective effects [52]. 

We determined the set of mitochondrial proteins that interact with Mitofilin using an immunoprecipitation and mass spectrometry approach. We have identified four SLC25A subunits (3, 5, 11, and 22) as proteins of interest (Figure 7). SLC25A3 promotes the transport of phosphate into the mitochondrial matrix, either by proton cotransport or in exchange for hydroxyl ions [53]. SLC25A5, also referred to as ADP/ATP translocase 2, translocates ADP from the cytosol into the mitochondrial matrix while transporting ATP in the opposite way [54]. SLC25A11 is the mitochondrial 2-oxoglutarate/malate carrier protein [55], while SLC25A22 is one of the two mitochondrial glutamate/H+ symporters [56]. The analysis of the function of these four SLC25As indicates that they are involved in the functioning of the electron transport chain (ETC), suggesting that dysregulation of these SLC25As solute carriers might result in increased ROS production (Figure 5B). Interestingly, compared to WT, we found that Mitofilin^+/−^ mice display a pronounced dysregulation of these four SLCA25s (Figure 8), which is associated with increased ROS production. Our data indicate that loss of the structural protein, Mitofilin, disrupts its interaction with SLC25A carriers resulting in their dysfunction that disturbs the ETC function leading to increased ROS production. These observations are in line with the loss of Mitofilin impacting ROS production that favors the release of mtDNA in the cytosol. Several studies have indicated that mitochondrial ROS production favors mtDNA mutations and damage [57,58], and importantly, I/R injury increases the damage of cardiomyocyte membranes and subcellular structures, including mtDNA) [59]. In normal conditions, mtDNA is stored within the mitochondria in response to stress; it can be released into the cytosol, where it encounters cytosolic DNA sensors. We recently revealed that Mitofilin reduction after acute kidney injury (AKI) promotes an increase in the release of mtDNA in the cytosol, where it favors the activation of the cGAS/STING pathway leading to enhanced phosphorylation of p65, known to translocate into the nucleus and initiate protein transcription [50]. In the cytosol, mtDNA can trigger various pro-inflammatory signaling pathways, including endosomal localized TLR9, cytosolic cGAS-STING axis, or cytosolic inflammasome (AIM2 or NLRP3) [60]. Interestingly, Mitofilin+/− mice exhibited increased transcription of pro-inflammatory markers, including TNF-α, ICAM, and IL-6 after I/R (Figure 6B–D). Together, these results support the hypothesis that Mitofilin downregulation promotes mtDNA release in the cytosol and activates the cGAS/STING/p-P65 pathway that favors transcription of pro-inflammatory markers, which exacerbate I/R injury. This funding is supported by the observation that pathological stimulation of retinal microvascular endothelial cells induces mtDNA escape into the cytosol that is recognized by the DNA sensor cGAS that enhances the expression of inflammatory markers via the STING/TBK1 axis [61].

## 5. Conclusions

We report that the knockdown of Mitofilin in mice increases mitochondrial structural damage and dysfunction, which results in critical failure of mitochondria to regulate Ca^2+^ homeostasis leading to increased mitochondrial sensitivity to Ca^2+^ overload that promotes mPTP opening and, subsequently, causes cardiomyocyte death. Conversely, loss of Mitofilin induces dysregulation of SLC25As solute carriers that promote an increase in ROS generation that facilitates the release of mtDNA release into the cytosol, where it activates the signaling pathway that leads to augmented transcription of a nuclear transcription of pro-inflammatory cytokines that subsequently exacerbates I/R injury (Figure 10). 

## Figures and Tables

**Figure 1 antioxidants-12-00921-f001:**
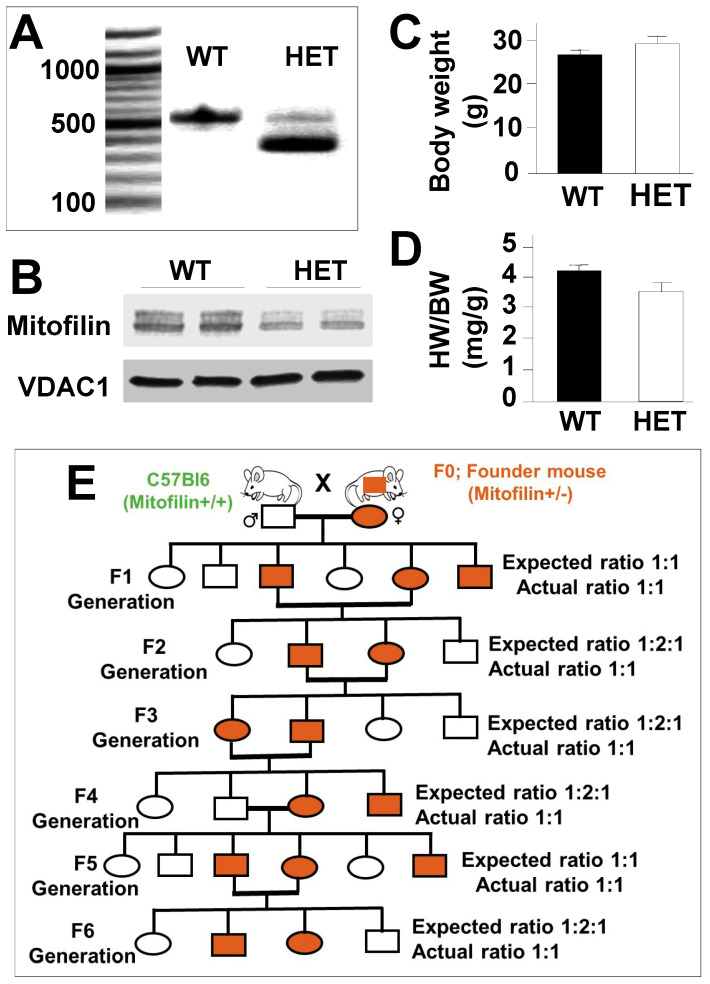
Full-body Mitofilin^−/−^ mice do not survive after multiple crossings of Mitofilin^+/−^ (HET) mice. (**A**). Genotype analysis was used to determine the Mitofilin^+/−^ (HET) mice versus the littermate WT mouse. Mouse genotype was obtained using RT-qPCR analysis with a specific primer, as shown in Table 1. Note that Mitofilin^+/−^ heterozygote mice display both alleles (knockout and WT bands). (**B**). Immunoblot confirming the reduction in Mitofilin levels in Mitofilin^+/−^ mice versus littermate WT mice. (**C**). Graph showing the body weight and the ratio heart weight/body weight (HW/BW) (**D**) in WT and Mitofilin^+/−^ mice indicating no significant difference in the ratio HW/BW between both groups. Values are expressed as mean ± SEM; *n* = 6/group. (**E**). Mitofilin genealogical tree illustrating that full-body Mitofilin^−/−^ (homozygote) mice do not survive after several crossing Mitofilin^+/−^ mice.

**Figure 2 antioxidants-12-00921-f002:**
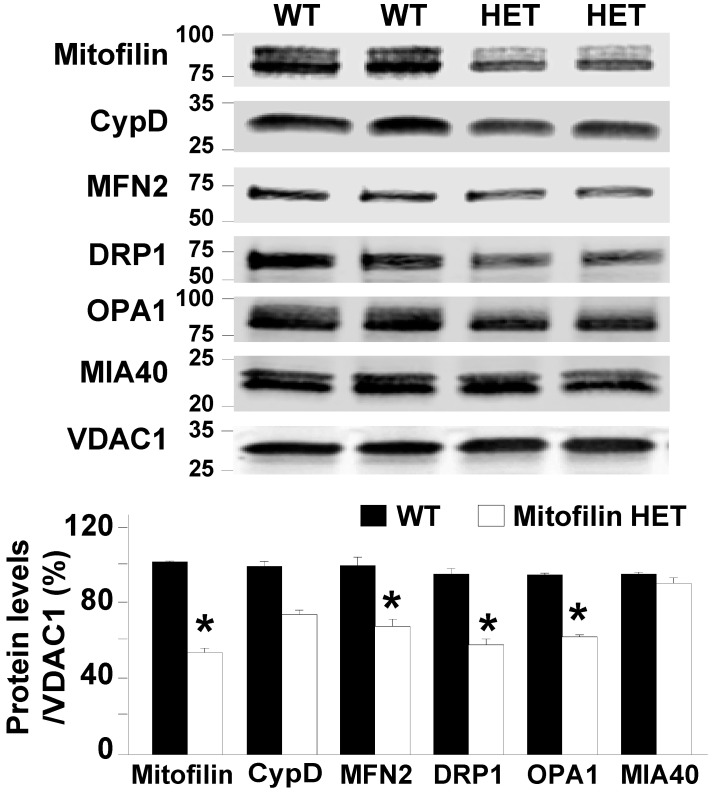
Mitofilin^+/−^ heart Mitochondria exhibit impaired mitochondrial dynamics in normal conditions. Immunoblots and corresponding bar graphs show a decrease in the levels of MFN2, DRP1, and OPA1 in Mitofilin^+/−^ versus WT mice in normal conditions. Note that the levels of Cyclophilin D and MIA40 (CHCHD4) were only slightly reduced in Mitofilin^+/−^ versus WT mice. Values are expressed as mean ± SEM; * *p* < 0.05 versus. WT sham group (*n* = 4/group).

**Figure 3 antioxidants-12-00921-f003:**
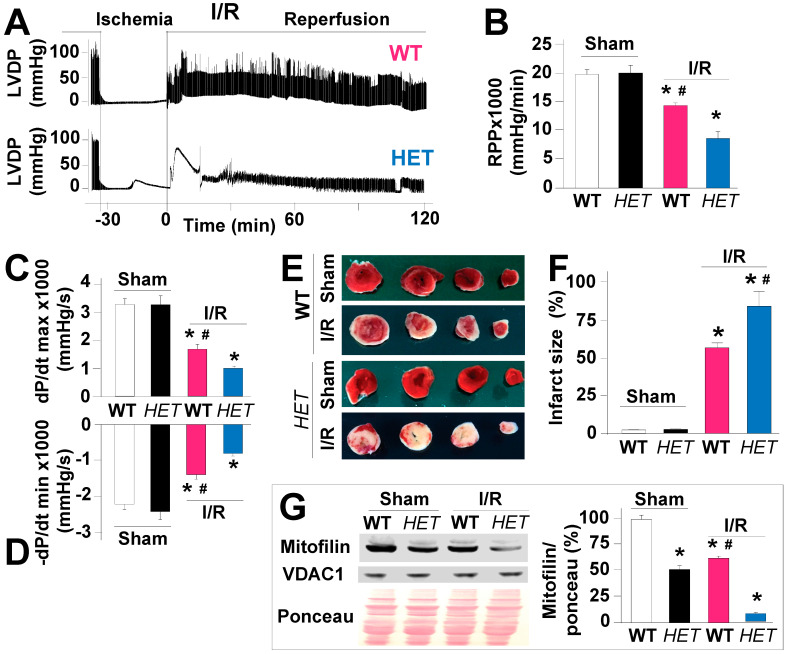
Mitofilin^+/−^ mice display reduced cardiac function recovery and increased myocardial infarct size after I/R. (**A**). Recordings of the left ventricular developed pressure (LVDP) of the heart function isolated from WT and in Mitofilin^+/−^ mice. Traces show the cardiac function from pre-ischemia to the end of 120 min reperfusion. Note that the cardiac functional recovery during reperfusion was decreased in Mitofilin^+/−^ hearts (blue) as compared to WT (pink). (**B**). Graph showing the decrease in cardiac functional % Recovery in Mitofilin^+/−^ heart compared to WT, as measured by the rate pressure product (RPP) at 120 min reperfusion and the maximal rate of rise of left ventricular pressure (dP/dt max, in (**C**)) and the maximum isovolumetric rate of relaxation (−dP/dt min, in (**D**)). (**E**). Representative images showing the reduction in myocardial infarct size (IS) in Mitofilin^+/−^ mice versus littermate WT mice. (**F**). Bar graph showing an increase in myocardial infarct size in Mitofilin^+/−^ hearts compared to WT hearts. (**G**). Immunoblot showing an excessive reduction in Mitofilin levels in Mitofilin^+/−^ mice versus littermate WT mice after. Values are mean ± SEM, * *p* < 0.05 versus. WT sham group, # *p*< 0.05 WT-I/R versus Mitofilin^+/−^ mice, *n* = 6/group.

**Figure 4 antioxidants-12-00921-f004:**
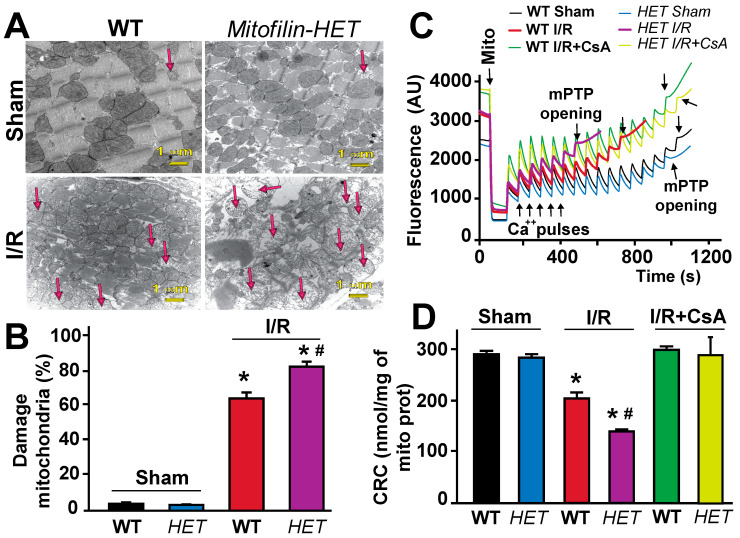
Mitofilin^+/−^ mice display an increase in mitochondrial damage and a reduced calcium retention capacity after I/R. (**A**). Images of mitochondria in cardiac tissues obtained using electron microscopy show a dramatic increase in cristae disruption after I/R compared to sham in WT mice. However, in Mitofilin^+/−^ mitochondria, the levels of cristae disruption were increased when compared to littermate WT mice after I/R. There was no difference but a slight increase in the levels of cristae disruption in both WT and Mitofilin^+/−^ mice sham mice. (**B**). Bar graph showing the percentage of damaged mitochondria in each group. Fragmented or disrupted cristae with empty spaces (in the matrix) were considered damaged mitochondria, while mitochondria with dense continuous cristae were considered as good or undamaged. A minimum of 100 mitochondria were counted in each group. Values are expressed as mean ± SEM; * *p* < 0.05 versus. WT sham group, # *p* < 0.05 versus. WT-I/R group (*n* = 5/group). (**C**). Mitofilin^+/−^ mice exhibit increased mitochondrial Ca^2+^ retention capacity (CRC) required to induce the mitochondrial permeability transition pore (mPTP) opening after I/R injury. Typical spectrofluorometric recordings of Ca^2+^ overload in mitochondria isolated from hearts after 35 min ischemia followed by 6h reperfusion. Subsequent 20 nmol-mg−1 of protein Ca^2+^ pulses were delivered until a spontaneous massive release was observed, presumably to the opening of mPTP (arrows). (**D**). The graph shows no difference in mitochondrial CRC in WT and Mitofilin^+/−^ mice mitochondria from sham and after I/R supplemented with cyclosporine A (mPTP opening inhibitor, 2 μM) in both groups. Values are expressed as mean ± SEM; * *p* < 0.05 versus. WT sham group, # *p* < 0.05 WT-I/R versus Mitofilin^+/−^ mice; *n* = 5/group.

**Figure 5 antioxidants-12-00921-f005:**
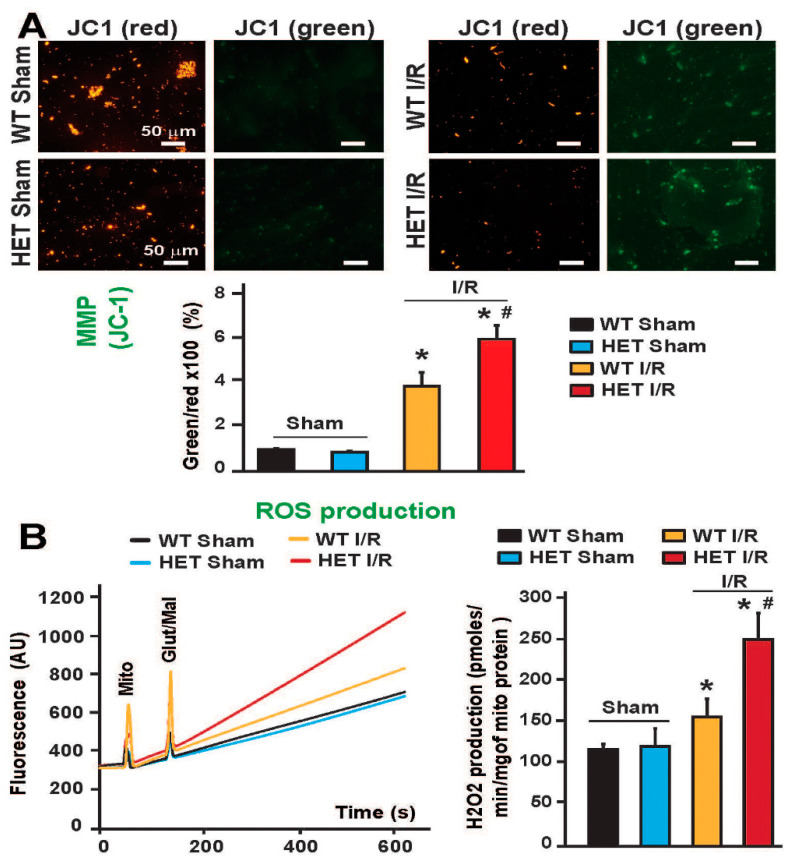
Mitochondria from Mitofilin^+/−^ mice are more uncoupled and generate more ROS following I/R. (**A**). Images of mitochondria and graph show an increase in the green/red fluorescence ratio of JC-1 dye in Mitofilin^+/−^ mice mitochondria compared to WT mitochondria. Values are as mean ± S.E.M, *n* = 4/group. * *p* < 0.05 vs vehicle, # *p* < 0.05 WT-I/R versus Mitofilin^+/−^ mice. (**B**). ROS produced on the complex I was measured in mitochondria from WT and Mitofilin^+/−^ heart. In normal conditions (non-ischemic), both groups produced similar levels of ROS (118 ± 6 pmoles/min/mg of mitochondrial protein in WT sham versus 123 ± 21 pmoles/min/mg of mitochondrial protein in Mitofilin^+/−^ sham mitochondria). However, after the ischemic insult, Mitofilin^+/−^ mitochondria generated more ROS than WT mitochondria (152 ± 17 pmoles/min/mg of mitochondrial protein in WT mitochondria versus 253 ± 232 pmoles/min/mg of mitochondrial protein in Mitofilin^+/−^ mitochondria; (**B**). When taken together, these findings suggest that Mitofilin knockdown increases the production of mitochondrial pro-deleterious ROS after I/R.

**Figure 6 antioxidants-12-00921-f006:**
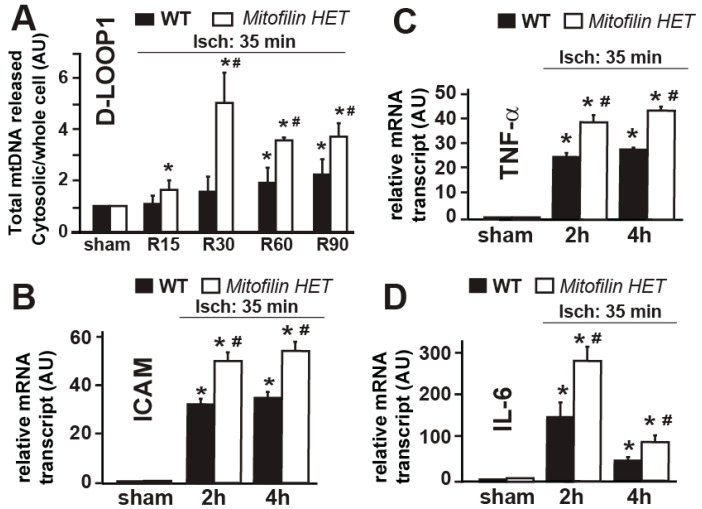
Mitofilin^+/−^ mice exhibit increased mtDNA release in the cytosol and decreased inflammation after I/R. (**A**). The graph shows an increase in the release of mtDNA (D-LOOP) in the cytosol in WT and Mitofilin^+/−^ mice in hearts subjected to 30 min ischemia followed by 30, 60, and 90 min reperfusion compared to respective sham hearts. However, in Mitofilin^+/−^ mice, the release of mtDNA in the cytosol was much higher when compared to littermate WT mice after I/R as earlier as 15 min reperfusion. Note that, in WT as well as Mitofilin^+/−^ sham mice, the level of mtDNA release in the cytosol was not different between these groups. Values are expressed as mean ± SEM; * *p* < 0.05 versus. WT sham group, # *p* < 0.05 versus. WT-I/R, (*n* = 4/group). (**B**–**D**). Graphs show an increase in the levels of pro-inflammatory cytokines, IL-6, ICAM-1, and TNF-6 in WT and Mitofilin^+/−^ hearts after 35 min reperfusion followed by 2 and 4 h reperfusion compared to respective sham hearts. Nevertheless, in Mitofilin^+/−^ hearts, the release of these inflammatory markers was much higher when compared to littermate WT mice after I/R. In both WT and Mitofilin^+/−^ sham mice, there was no difference in these pro-inflammatory cytokines. Values are expressed as mean ± SEM; * *p* < 0.05 versus. WT sham group, # *p* < 0.05 versus. WT-I/R, (*n* = 6/group).

**Figure 7 antioxidants-12-00921-f007:**
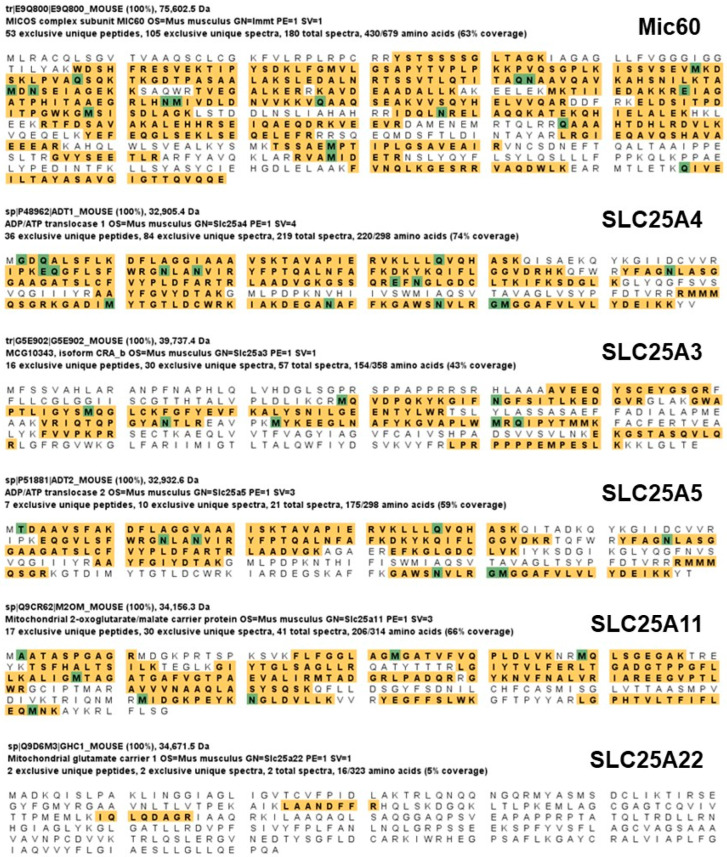
Mitofilin interacts with several SLC25As solute carriers. Sequence coverage maps from Scaffold 5 (Proteome Software) processing of the mass spectrometry data after database searching by Mascot (Matrix Science) of the UniProt_Mouse database combined with a database of common contaminants. Sequences highlighted in gold correspond to assignments made with the following confidence cutoff settings: protein threshold, 99%; peptide threshold, 95%; minimum number of peptides. Residues highlighted in green represent modifications generated during sample digestion: M, methionine oxidation; N and Q, deamidation; E, pyroglutamate. [Note that there were multiple repeats of many spectra; only a small percentage of the green highlighted residues were modified]. Note that mass spectrometry analyses were performed at the University of Texas Health Science Center at San Antonio (UTHSCSA) Core Facility.

**Figure 8 antioxidants-12-00921-f008:**
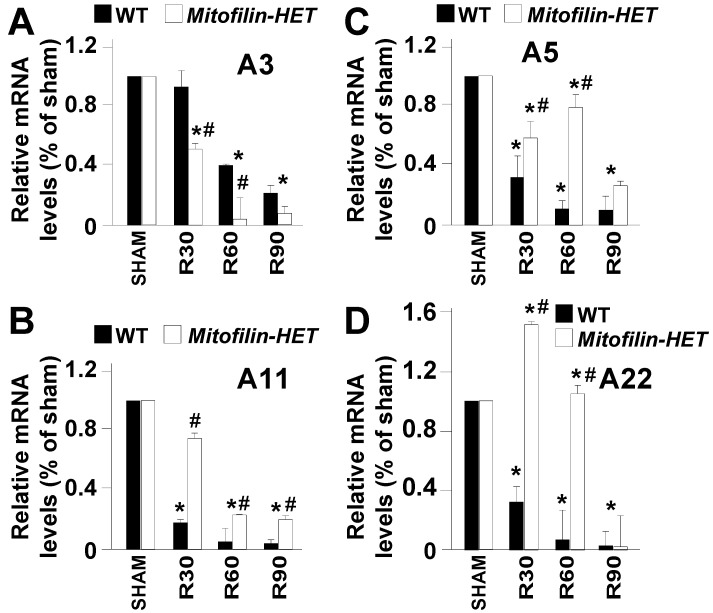
Mitofilin^+/−^ mice display dysregulated SLC25As solute carrier function after I/R. (**A**–**D**), Graphs showing dysregulation of the SLC25As solute carriers in Mitofilin^+/−^ mice compared to littermates WT after 35 min ischemia followed by reperfusion for 30, 60, and 90 min. The levels of SLC25A3 were progressively reduced (**A**), while the levels of SLC25A 5, 11, and 22 were increased (**C**,**D**) in Mitofilin^+/−^ mitochondria compared to the respective WT group. Values plotted were obtained from RT-qPCR analysis with a specific primer, as shown in Table 1. Values are expressed as mean ± SEM; * *p* < 0.05 versus. WT sham group, # *p* < 0.05 versus. WT-I/R, (*n* = 3/group).

**Figure 9 antioxidants-12-00921-f009:**
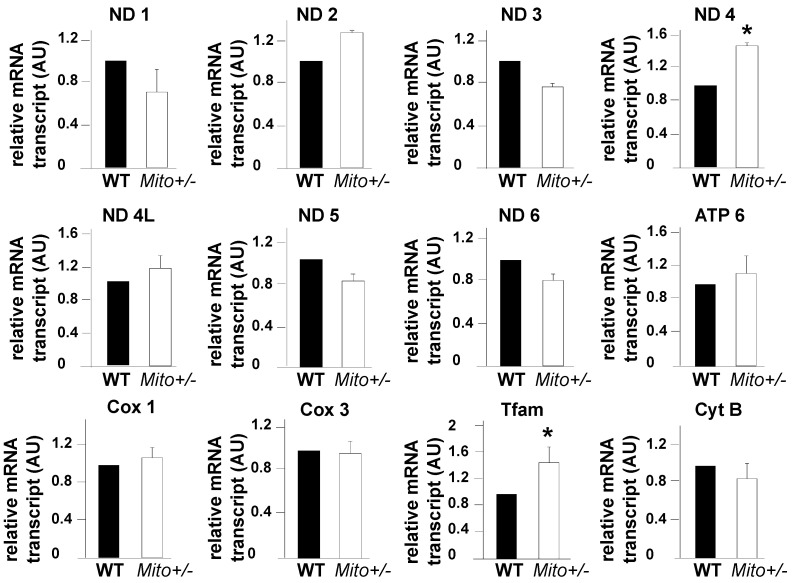
Mitofilin downregulation does not affect the mitochondrial transcription of most of the proteins. Graph showing the expression levels of the 13 transcribed in mitochondria. Values plotted were obtained from RT-qPCR analysis with a specific primer, as shown in Table 1. Values are expressed as mean ± SEM; * *p* < 0.05 versus *n* = 3/group.

**Figure 10 antioxidants-12-00921-f010:**
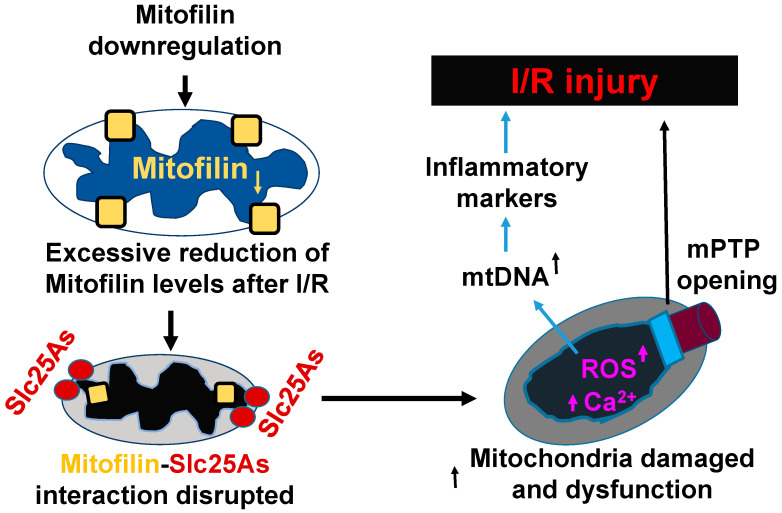
Graphic abstract summarizing the proposed mechanism depicted whether the Knockdown of Mitofilin increases myocardial injury and inflammation after I/R. Knockdown of Mitofilin in mice increases mitochondrial structures damage and dysfunction, which results in critical failure of mitochondria to regulate Ca^2+^ homeostasis leading to increased mitochondrial sensitivity to Ca^2+^ overload that promotes mPTP opening and, subsequently, causes cardiomyocyte death. Conversely, loss of Mitofilin induces dysregulation of SLC25As solute carriers that promote an increase in ROS generation, which is proposed to facilitate the release of mtDNA into the cytosol, which activates signaling pathways that amplifies the nuclear transcription of pro-inflammatory markers that subsequently exacerbate I/R injury.

**Table 1 antioxidants-12-00921-t001:** List of antibodies used in this study.

Antibody Target	Supplier	Catalog	Concentration
Mitofilin/Mic60	Proteintech Group	10179-1-AP	1 µg/mL
MIA40	Proteintech Group	21090-1-AP	1 µg/mL
OPA1	Proteintech Group	27733-1-AP	1 µg/mL
MFN2	Cell Signaling Technology	9482S	1 µg/mL
CYPD	Thermo Fisher	455900	1 µg/mL
VDAC1	EMD	MABN504	1 µg/mL
DRP1	Cell Signaling Technology	5391s	1ug/ml
IRDye 800CW Goat anti-Rabbit	LI-COR	926-32211	0.1 µg/mL
IRDye 680RD Goat anti-Mouse	LI-COR	926-68070	0.1 µg/mL

**Table 2 antioxidants-12-00921-t002:** List of forward and reverse primers used in this study.

Name	Primer Target	Sequence 5′ TO 3′	Supplier
ATP6	ATP6 CV f	TCCCAATCGTTGTAGCCATCA	Eurofins Genomics LLC
	ATP6 CV r	AGACGGTTGTTGATTAGGCGT	
COX 1	COX 1 f	ATCACTACCAGTGCTAGCCG	Eurofins Genomics LLC
	COX 1 r	CCTCCAGCGGGATCAAAGAA	
COX 2	COX 2 f	ACCTGGTGAACTACGACTGCT	Eurofins Genomics LLC
	COX 2 r	TCCTAGGGAGGGGACTGCTC	
COX 3	COX 3 f	CCAAGGCCACCACACTCCTA	Eurofins Genomics LLC
	COX 3 r	GGTCAGCAGCCTCCTAGATCA	
CYTB	CYTB f	GGCTACGTCCTTCCATGAGG	Eurofins Genomics LLC
	CYTB r	TGGGATGGCTGATAGGAGGT	
ND 1	ND 1 f	CTAGCAGAAACAAACCGGGC	Eurofins Genomics LLC
	ND 1 r	CCGGCTGCGTATTCTACGTT	
ND 2	ND 2 f	CCTCCTGGCCATCGTACTCA	Eurofins Genomics LLC
	ND 2 r	GAATGGGGCGAGGCCTAGTT	
ND 3	ND 3 f	TAGTTGCATTCTGACTCCCCCA	Eurofins Genomics LLC
	ND 3 r	GAGAATGGTAGACGTGCAGAGC	
ND 4	ND 4 f	CGCCTACTCCTCAGTTAGCCA	Eurofins Genomics LLC
	ND 4 r	TGATGTGAGGCCATGTGCGA	
ND 4L	ND 4L f	AGCTCCATACCAATCCCCATCAC	Eurofins Genomics LLC
	ND 4L r	GGACGTAATCTGTTCCGTACGTGT	
ND 5	ND 5 f	GGCCCTACACCAGTTTCAGC	Eurofins Genomics LLC
	ND 5 r	AGGGCTCCGAGGCAAAGTAT	
ND 6	ND 6 f	CTTGATGGTTTGGGAGATTGG	Eurofins Genomics LLC
	ND 6 r	ACCCGCAAACAAAGATCACC	
TFAM	TFAM f	GCCCGGCAGAGACGGTTAAA	Eurofins Genomics LLC
	TFAM r	GCCGAATCATCCTTTGCCTCC	
Slc25A3	Slc25a3 f	TGACATTTGTGGCAGGTTACA	Eurofins Genomics LLC
	Slc25a3 r	AGTCAGCAGGGTGGGAGAC	
Slc25A5	Slc25a5 f	GATGCCGCTGTGTCCTTC	Eurofins Genomics LLC
	Slc25a5 r	TATCTGCCGTGATTTGCTTG	
Slc25A11	Slc25a11 f	CCCGTACCTCCCCTAAGTCT	Eurofins Genomics LLC
	Slc25a11 r	AACTGCATCCGGTTCTTCAC	
Slc25A13	Slc25a13 f	TCCCACTTTTGGCAGAGATT	Eurofins Genomics LLC
	Slc25a13 r	CGGATTTTCACAATCTCTAAAGG	
Slc25A20	Slc25a20 f	GGTGTGTTCACCACAGGAATCA	Eurofins Genomics LLC
	Slc25a20 r	CCCCTGAAGAAGCCTGAAT	
Slc25A22	Slc25a22 f	TGCTTGAGGTCTTTGTGTGC	Eurofins Genomics LLC
	Slc25a22 r	CTTATGGGTCCCCATCCCTA	
Slc25A42	Slc25a42 f	AGCAGGTTGCACCATGTAGA	Eurofins Genomics LLC
	Slc25a42 r	GAAGATCAGGGACCCAGGAC	
D-LOOP	D-LOOP f	CCAGTCTTAAACCGGAGA	Eurofins Genomics LLC
	D-LOOP r	CTATCACCCTATTAACCACTC	
TNF-α	TNF-α f	GAGAAAGTCAACCTCCTCTCTG	Eurofins Genomics LLC
	TNF-α r	GAAGACTCCTCCCAGGTATATG	
IL-6	IL-6 f	TAGTCCTTCCTACCCCAATTTCC	Eurofins Genomics LLC
	IL-6 r	TTGGTCCTTAGCCACTCCTTC	
ICAM-1	ICAM-1 f	GTGATGCTCAGGTATCCATCCA	Eurofins Genomics LLC
	ICAM-1 r	CACAGTTCTCAAAGCACAGCG	
18S	18S f	CGGCTACCACATCCAAGGAA	Eurofins Genomics LLC
	18S r	GCTGGAATTACCGCGGCT	
Mic60	Mic60 f	TAAGCAGTACCGCCATGTCTTCTGTCAAGTTATGGCC	Eurofins Genomics LLC
	Mic60 r	TGCTTAGCGGCCGCACGCGTCTTGTGGAAGGGC	

## Data Availability

Not applicable.

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
