# Peer review of "Mitofilin Heterozygote Mice Display an Increase in Myocardial Injury and Inflammation after Ischemia/Reperfusion"

_antioxidants, 2023, doi:10.3390/antiox12040921_

Round 1

Reviewer 1 Report

In this manuscript, Feng et al. show how the downregulation of MIC60 (mitofilin) impacts heart damage by I/R. To address this question, authors use a novel mouse transgenic model MIC60+/-since the +/+ appears lethal. Authors observe the main changes in the MIC60+/- hearts after I/R, which increase infartc area. Moreover, authors describe a serie of mitochondria alterations that appear in MIC60+/- only after I/R: a slight decrease in profusion and fission proteins, lower CRC than controls but entirely rescued by treatment with CsA, altered mitochondrial ultrastructure, lower membrane potential, higher ROS production, and higher mtDNA release with the production of pro-inflammatory markers. Moreover, based on previously published pulldown experiments showing SLC25As solute carriers as putative interactors of MIC60, authors measure transcripts for the different SLC25As carriers change, which relevance for the identified phenotype is not investigated.

The novelty of the research is the use of a transgenic mice model for MIC60, being this paper the first time the model has been published. For this reason, a more detailed characterization should be included useful for the correct interpretation of the results. Moreover, the detailed M&M description of how the model was generated is lacking and must be included. The results shown along the manuscript are consistent, even though some controls are missing, and conclusions in some paragraphs are overstated (see specific points). 

I advise a major revision of the paper addressing the points listed here below.

1. Authors present for the first time a mouse model full-body KO for Mitofilin, and the only description in M&M is

 ……generated by pro-viral insertion of a recombinant retrovirus to interfere with (136) RNA expression of the IMMT gene as described in [12].

A full description of the genetic modification is required. What do authors mean with “interfere with RNA expression”?. Moreover, a basic phenotypical  characterization should be included: body weight during growth and heart phenotype at basal conditions (staining to observe tissue integrity and alterations, heart size, etc.) in the same time point used for the experiments,e.g. if mice were 4 months old, the characterization should be done at the same time point or up to the same time point. This characterization is fundamental to understanding the response observed after I/R. Indeed, the TEM images show a heart tissue disruption at the moment of the experiments, that could impact the I/R response.

Moreover, it could be beneficial to show at which stage there is embryonic lethality for homozygous mice.

2. The authors want to determine whether Mitofilin downregulation affects mitochondria dynamics in normal conditions. To do so, they perform Western blot analysis of some mitochondrial proteins. Among these, the profusion MFN2 and OPA1 and the fission DRP1 protein, observing a slight decrease in the protein levels. Based on this result they conclude: These observations 298 suggest that Mitofilin knockdown affect mitochondrial dynamics that results in a slightly 299 damaged mitochondria in normal conditions.

This result and conclusion are wrongly assembled. First, the authors show a slight decrease in fusion ad fission proteins, but there is no measurement of mitochondrial dynamics or damage and there is no indication that this slight decrease can induce alterations. Indeed, mitochondria appear normal and functional in these conditions. Thus, the conclusion is not supported by the data. Second, the authors use lysates from isolated mitochondria, but DRP1 is a cytosolic protein that translocates to mitochondria to induce fission after post-traslational modifications. The presence of DRP1 in both WT and MIC60+/- isolated mitochondria denotes contamination or any other problem that must be resolved.

A decrease in mitochondrial mass could also explain the slight decrease in IMM proteins. This possibility should also be addressed in cardiomyocytes from these mice using proper approaches.

3. Authors write: ....Note that after I/R, the levels (331) of Mitofilin loss were more pronounced  in HET mitochondria compared to littermates (332) WT (Fig. 3G).These results suggest an important role for Mitofilin in the initiation and 333 development of the cardiac response to I/R injury.

Even if it is clear that MIC60 +/- mice hearts are more sensitive to I/R damage, it is unclear why the authors claim that Mitofilin is important for the initiation and development of cardiac response to I/R. How can authors distinguish the initiation event from the development?. The fact that there is a further loss of MIC60 in mitochondria after I/R could also be a consequence of the already altered heart tissue well visible in Fig 4A. Are other IMM proteins, such as OPA1 and Chchd3, altered? Last, mitochondrial and cytosolic protein loading control should be included to exclude that the reduction is due to loss of mitochondrial mass.

The panel 3G is not described in the figure caption

4. Authors write: We assessed the disruption of the cristae morphology in normal and after I/R con-352 ditions in both WT and Mitofilin+/- mice. In non-ischemic hearts, we found that cristae 353 morphology in Mitofilin+/- mice was slightly disorganized, but but not disrupted when com-354 pared to WT . However, following I/R, Mitochondria 355 from Mitofilin+/- mice hearts exhibited more damaged cristae morphology versus WT

Looking a the TEM images in basal conditions, it is obvious that heart tissue integrity is lost. However, mitochondria ultrastructure appears well-mantained. Indeed, the CRC analysis shows no differences between WT and MIC60+/- mitochondria in control conditions. This is further confirmed in Fig.5A showing no differences in the membrane potential in the same conditions. MoreoverI/R phenotype is rescued by CsA. All these data support that mitochondria are functional and in shape in MIC60+/- hearts. Despite these results, authors describe the cristae disruption also in basal conditions. Can authors comment on this?. It would be helpful to show/specify  with higher magnification in the paper what authors define “disruption of the cristae morphology”. On the other hand, the TEM images after I/R clearly show a swollen mitochondrial phenotype in MIC60+/-  respect to WT, which is fully protected by CsA, further supporting that mitochondria are functional in MIC60+/- hearts. Whether and how this mitochondrial phenotype is influenced by the already disrupted tissue is hard to say and should be discussed in the paper.

5. Based on the TEM images, the authors investigate the membrane potential as an indicator of dysfunction using JC1 in WT and MIC60+/- isolated mitochondria after I/R. These experiments should be completed by including a panel with the mitochondria from CsA-I/R treated WT and MIC60+/- hearts, data that will also support the CRC results.

6. Authos write: Mitofilin+/- mice display dysregulated SLC25As solute carrier function after I/R.

Based in a pulldown experiment (previously published to evidence other MIC60 interctors), authors study how several SLC25A carriers are affected by MIC60 downregulation. To do so, transcripts level were measured in basal conditions and after I/R. The results show a decrease in SLC25A3 transcripts after reperfusion and an increase in the A5,A11 and A22. These results require to be studied in parallel with protein levels. Still, there is no results showing a functional dysregulation of these carriers, as claimed in the head of the paragraph. Moreover, also the conclusion is based in assumptions: Together these results indicate that the mechanism of Mitofilin knockdown 506 induced cardio-deleterious effect after I/R involve dysregulation of SLC25As solutes car-507 riers that might result in increased ROS production. To demosstrate this, genetic manipulation of this carries should be done and then measure ROS production.

It is not described if values are normalized to each Sham. Are the initial values different in WT and MIC60+/- ?

I would avoid to add a figure previously published (pulldown experiment Fig 7) even if the table shows some modifications to include specific proteins of interest for this investigation. Authors can recall the paper published and include a table with the spectra for each protein of interest.

7. Several M&M are lacking: ROS measurement, mice generation, primers for pro-imflammatory markers, etc.. please check.

Minor points

·       Some figure captions are too much simplified and imprecise. Some examples here below:

 Figure 3…..Typical recording of isolated perfused heart function…. Typical recording? It should be specified what is being recorded;

 …..Graph showing the decrease in cardiac functional % Recovery  of in Mitofilin+/- mice compared to WT…. I would specify in Mitofilin +/- hearts since the experiment is ex vivo

English better to be revised (typos and others). Some examples here below:

               -we only obtained either WT or Mitofilin+/- mice or Mitofilin+/+(WT).

               -hear weight and the ration body weight/body weight

               -the levels of Mitofilin was inverse proportional of 311 the extent of infarct size after I/R

·         Mouse proteins are designated with capital letters: e.g. MFN2 and not mfn2 (check others in the manuscript)

·         The panel 3G is not described in the figure caption

Author Response

Reviewer 1

We would like to thank the reviewer for all his constructive comments.

1. Authors present for the first time a mouse model full-body KO for Mitofilin, and the only description in M&M is

 ……generated by pro-viral insertion of a recombinant retrovirus to interfere with (136) RNA expression of the IMMT gene as described in [12].

A full description of the genetic modification is required. What do authors mean with “interfere with RNA expression”?. Moreover, a basic phenotypical  characterization should be included: body weight during growth and heart phenotype at basal conditions (staining to observe tissue integrity and alterations, heart size, etc.) in the same time point used for the experiments,e.g. if mice were 4 months old, the characterization should be done at the same time point or up to the same time point. This characterization is fundamental to understanding the response observed after I/R. Indeed, the TEM images show a heart tissue disruption at the moment of the experiments, that could impact the I/R response.

We agree with the reviewer’s comment regarding a full description of the genetic modification of our created mouse and we have added a paragraph explaining the genetic modifications in the materials and methods. Regarding the phenotype characterization of the Mitofilin heterozygote mouse we created, Figure 1A-D provides our initial characterization of this mouse model. Data provided include genotyping, Mitofilin protein levels, body weight and the ratio heart weight/bodyweight at adult age (4-6 month old). Note that all the assessments performed in normal conditions (basal) including heart function, myocardial infarction, mitochondria morphology and function, body weight as well as the ratio heart weight/bodyweight were not significantly different compared to littermate WTs. In addition, Mitofilin heterozygote mice live up to two years, which is not much different to littermate WTs. We noticed some tissue damage with (TEM) in Mitofilin HET; however, the overall mitochondria function in sham animal were not different compared to littermate WT, indicating a minor damage phenotype. Together, our data point to Mitofilin heterozygote mouse displaying a much similar phenotype like their littermate WTs in basal conditions, which is supported by the fact that they live  as long as their littermate WT. We will perform other characterization in pups and old animals in our future studies.

Moreover, it could be beneficial to show at which stage there is embryonic lethality for homozygous mice.

We agree with the reviewer’s comment to study the embryonic lethality for homozygous mice in our future studies. However, this will not add much to the overall impact of this manuscript, which is focused on whether Mitofilin heterozygote mice display an increase in myocardial injury and inflammation after ischemia/reperfusion. It is hard to understand how the embryonic lethality stage of homozygous mice will justify the weakness of Mitofilin heterozygote heart to ischemia/reperfusion injury. 

2. The authors want to determine whether Mitofilin downregulation affects mitochondria dynamics in normal conditions. To do so, they perform Western blot analysis of some mitochondrial proteins. Among these, the profusion MFN2 and OPA1 and the fission DRP1 protein, observing a slight decrease in the protein levels. Based on this result they conclude: These observations 298 suggest that Mitofilin knockdown affect mitochondrial dynamics that results in a slightly 299 damaged mitochondria in normal conditions.

This result and conclusion are wrongly assembled. First, the authors show a slight decrease in fusion ad fission proteins, but there is no measurement of mitochondrial dynamics or damage and there is no indication that this slight decrease can induce alterations. Indeed, mitochondria appear normal and functional in these conditions. Thus, the conclusion is not supported by the data. Second, the authors use lysates from isolated mitochondria, but DRP1 is a cytosolic protein that translocates to mitochondria to induce fission after post-traslational modifications. The presence of DRP1 in both WT and MIC60+/- isolated mitochondria denotes contamination or any other problem that must be resolved.

A decrease in mitochondrial mass could also explain the slight decrease in IMM proteins. This possibility should also be addressed in cardiomyocytes from these mice using proper approaches.

We also agree with the reviewer’s comment regarding the Figure 2. We have edited our conclusion accordingly. However, it is hard to explain the presence of DRP1 in mitochondria fraction by a cytosolic contamination. It is well known that DRP1 is a cytosolic protein that translocates to mitochondria to induce fission after post-translational modifications. Because fusion/fission are normal processes in cardiomyocytes, it therefore not excluded to detect DRP1 in the mitochondria fraction. The big size of the DRP1 band result from the increase in the membrane exposure time. Note that VDAC1 and CypD levels, which are specific to mitochondria, are not changed in Mitofilin heterozygote mice and we measured GAPDH in the Mitochondrial fraction and we did not get any bands. This observation associated with EM images represented in Figure 4 support the idea that the decrease in the protein levels presented in Figure 2 does not result from a reduction of mitochondrial mass. We postulate that the concomitance of the decrease in the levels of proteins involved in both fusion and fission justifies the absence of mitochondria damage observed in heterozygote mice. However, we agree with the pertinence of the reviewer’s observation.

3. Authors write: ....Note that after I/R, the levels (331) of Mitofilin loss were more pronounced  in HET mitochondria compared to littermates (332) WT (Fig. 3G).These results suggest an important role for Mitofilin in the initiation and 333 development of the cardiac response to I/R injury.

Even if it is clear that MIC60 +/- mice hearts are more sensitive to I/R damage, it is unclear why the authors claim that Mitofilin is important for the initiation and development of cardiac response to I/R. How can authors distinguish the initiation event from the development?. The fact that there is a further loss of MIC60 in mitochondria after I/R could also be a consequence of the already altered heart tissue well visible in Fig 4A. Are other IMM proteins, such as OPA1 and Chchd3, altered? Last, mitochondrial and cytosolic protein loading control should be included to exclude that the reduction is due to loss of mitochondrial mass.

The panel 3G is not described in the figure caption

We agree with the reviewer’s comment and have edited the manuscript accordingly. Note that in Mitofilin heterozygote mitochondria, MIA40 (called CHCHD4 in human cells), CypD and VDAC1 protein levels were not significantly different when compared to littermate WTs. The immunoblots data indicate that the reduction in MFN2, DRP1, and OPA1 observed in Mitofilin heterozygote mitochondria cannot be explain by the loss of mitochondrial mass. In addition, analysis of TEM images of mitochondria did not also suggest any reduction of the mitochondrial mass.

4. Authors write: We assessed the disruption of the cristae morphology in normal and after I/R con-352 ditions in both WT and Mitofilin+/- mice. In non-ischemic hearts, we found that cristae 353 morphology in Mitofilin+/- mice was slightly disorganized, but but not disrupted when com-354 pared to WT . However, following I/R, Mitochondria 355 from Mitofilin+/- mice hearts exhibited more damaged cristae morphology versus WT

Looking a the TEM images in basal conditions, it is obvious that heart tissue integrity is lost. However, mitochondria ultrastructure appears well-mantained. Indeed, the CRC analysis shows no differences between WT and MIC60+/- mitochondria in control conditions. This is further confirmed in Fig.5A showing no differences in the membrane potential in the same conditions. MoreoverI/R phenotype is rescued by CsA. All these data support that mitochondria are functional and in shape in MIC60+/- hearts. Despite these results, authors describe the cristae disruption also in basal conditions. Can authors comment on this?. 

As shown in Figure 5, there is no difference between both basal groups in regard to mitochondria morphology and function. Our analysis in the manuscript regarding the cristae disruption in basal conditions is based on the role of Mitofilin in maintaining cristae morphology. We report that in mice, full-body deletion of only one Mitofilin allele does not induce significant damage. This observation on mitochondria structure is compared with the increase in mitochondria damage associated with excessive reduction of Mitofilin caused by I/R insult, which causes degradation and loss of both Mitofilin alleles. 

It would be helpful to show/specify with higher magnification in the paper what authors define “disruption of the cristae morphology”. On the other hand, the TEM images after I/R clearly show a swollen mitochondrial phenotype in MIC60+/-  respect to WT, which is fully protected by CsA, further supporting that mitochondria are functional in MIC60+/- hearts. Whether and how this mitochondrial phenotype is influenced by the already disrupted tissue is hard to say and should be discussed in the paper.

We agree with the reviewer’s comment to define the concept of “disruption of the cristae morphology”. We have added a small section accordingly. Fragmented or disrupted cristae with empty spaces (in the matrix) were considered damaged mitochondria (368-370). Our hypothesis is that extensive reduction in Mitofilin levels after I/R in Mitofilin heterozygote mitochondria disrupts the cristae morphology via dysfunction of MICOS system resulting in the alteration of mitochondria function. 

5. Based on the TEM images, the authors investigate the membrane potential as an indicator of dysfunction using JC1 in WT and MIC60+/- isolated mitochondria after I/R. These experiments should be completed by including a panel with the mitochondria from CsA-I/R treated WT and MIC60+/- hearts, data that will also support the CRC results.

We do not really see whether adding data on CsA-treated in WT and MIC60+/- hearts after I/R to measure mitochondrial membrane potential would strengthen the current manuscript since there is a panel of experiments on CRC and CsA. In addition, the overall role of CsA in mitochondria membrane potential is well known.

6. Authos write: Mitofilin+/- mice display dysregulated SLC25As solute carrier function after I/R.

Based in a pulldown experiment (previously published to evidence other MIC60 interctors), authors study how several SLC25A carriers are affected by MIC60 downregulation. To do so, transcripts level were measured in basal conditions and after I/R. The results show a decrease in SLC25A3 transcripts after reperfusion and an increase in the A5, A11 and A22. These results require to be studied in parallel with protein levels. Still, there is no results showing a functional dysregulation of these carriers, as claimed in the head of the paragraph. Moreover, also the conclusion is based in assumptions: Together these results indicate that the mechanism of Mitofilin knockdown 506 induced cardio-deleterious effect after I/R involve dysregulation of SLC25As solutes car-507 riers that might result in increased ROS production. To demosstrate this, genetic manipulation of this carries should be done and then measure ROS production.

It is not described if values are normalized to each Sham. Are the initial values different in WT and MIC60+/- ?

 We agree that genetic manipulation of SLC25A carriers would be the best approach to determine the involvement of each of them in the mechanism responsible for the increase in I/R injury in Mitofilin. Mass spectrometry after immunoprecipitation with anti-Mitofilin antibody pulled down several SLC25A solute carriers. We studied whether reduction of Mitofilin after I/R can affect the trafficking of these carriers that are involved in the electron transfer chain function. We interestingly found the dysregulation of carriers (mRNA) in Mitofilin heterozygote mice. We agree that the levels of mRNA do not systematically indicate the protein levels. However, it is hard to also believe that change in mRNA transcript can be associated with an opposite level of a protein translation. Therefore, we did anticipate a similar change in the protein levels of these solute carriers in Mitofilin heterozygote mice. Values plotted were normalized to each Sham. We have edited the figure and legends accordingly.

I would avoid to add a figure previously published (pulldown experiment Fig 7) even if the table shows some modifications to include specific proteins of interest for this investigation. Authors can recall the paper published and include a table with the spectra for each protein of interest.

We agree with the reviewer’s comment and have edited the manuscript accordingly. The Figure 8 has been changed to focus on the interactions between Mitofilin and the different SLC25A subunits after IP with anti-Mitofilin antibody.

7. Several M&M are lacking: ROS measurement, mice generation, primers for pro-imflammatory markers, etc.. please check.

We agree with the reviewer’s comment. We have included a material and method section for the missing experiments.

Minor points

·       Some figure captions are too much simplified and imprecise. Some examples here below:

 Figure 3…..Typical recording of isolated perfused heart function…. Typical recording? It should be specified what is being recorded;

We agree with the reviewer’s comment and have edited the manuscript accordingly.

 …..Graph showing the decrease in cardiac functional % Recovery  of in Mitofilin+/- mice compared to WT…. I would specify in Mitofilin +/- hearts since the experiment is ex vivo

We agree with the reviewer’s comment and have edited the manuscript accordingly.

English better to be revised (typos and others). Some examples here below:

               -we only obtained either WT or Mitofilin+/- mice or Mitofilin+/+(WT).

               -hear weight and the ration body weight/body weight

               -the levels of Mitofilin was inverse proportional of 311 the extent of infarct size after I/R

We agree with the reviewer’s comment and have edited the manuscript accordingly.

·         Mouse proteins are designated with capital letters: e.g. MFN2 and not mfn2 (check others in the manuscript)

 We agree with the reviewer’s comment and have edited the manuscript accordingly.

·         The panel 3G is not described in the figure caption

We agree with the reviewer’s comment and have edited the manuscript accordingly.

Reviewer 2 Report

Feng et al. explored the impact of mitochondrial inner membrane protein (Mitofilin) in an experimental infarction model using the I/R technique.  While homozygous KO mice are lethal, heterozygous mice show a phenotype after I/R. Thus, mitofilin+/- mice show larger infarcts, greater mitochondrial structural damage, more mtDNA release and more ROS production, etc. after I/R.

Comments:

- The entire article contains numerous unnecessary spaces.

- Line 17: 'intact mice' certainly means wild-type mice, which is how it should be written.

- The introduction is written in great detail. Many passages can be shortened or moved to the discussion.

- The list of primers takes up an entire page and could be shown as supplemental material. The indication of the manufacturer seems a bit strange, as it is always the same one...

- Line 262: 'trials' is maybe not the best here… experiments would be more appropriate.

- Figure 1A: Please indicate that PCR was used here.

- Figure1C: Y-axis is not correctly labeled, not body weight is shown.

- Figure 1E: Supplement would be more appropriate for this information.

- In general, the figures are very large and could easily be reduced in size to save space.

- The legends contain far too much information, moreover the findings are described in detail and even papers are cited... This does not belong here and must be revised! The legend must show what was done, nothing more. Findings should be described in the text of the results.

- Figure 2: It is a bit confusing because the abbreviation for the proteins is used here and in the legend also the written version.

- Line 332: One ‘in’ should be deleted.

- Figure 3: Presumably after converting the graphs to PDF, display problems occurred.

- Figure 4C: The lines are difficult to distinguish... perhaps additional dashed or dotted lines could be used here.

- It is often not clear in the results how this was investigated. Isolated mitochondria from which tissue exactly? Infarct area?

- Figure 5: The order of the columns and the labeling is different, which is a bit confusing...

- Line 448: Fig. 8B?

- The graphic abstract in Figure 10 seems a bit unkind... you could do better!

Author Response

Reviewer 2

We would like to thank the reviewer for all his constructive comments.

Feng et al. explored the impact of mitochondrial inner membrane protein (Mitofilin) in an experimental infarction model using the I/R technique.  While homozygous KO mice are lethal, heterozygous mice show a phenotype after I/R. Thus, Mitofilin+/- mice show larger infarcts, greater mitochondrial structural damage, more mtDNA release and more ROS production, etc. after I/R.

Comments:

- The entire article contains numerous unnecessary spaces.

We agree with the reviewer’s comment. However, the proof will not have these empty spaces.

- Line 17: 'intact mice' certainly means wild-type mice, which is how it should be written.

We agree with the reviewer’s comment and have edited the manuscript accordingly.

- The introduction is written in great detail. Many passages can be shortened or moved to the discussion.

We don’t totally agree with the reviewer’s comment. As shown in the present manuscript proof, we did not consider to change the introduction and move some passage to the discussion. The main reason is to not completely change the structure of the manuscript that other reviewers have reviewed.

- The list of primers takes up an entire page and could be shown as supplemental material. The indication of the manufacturer seems a bit strange, as it is always the same one...

We agree with the reviewer’s comment. As presented in the present manuscript proof, it looks much nicer and well presented. We therefore did not consider putting that table in supplementary materials. We will discuss with the Editor if necessary.

- Line 262: 'trials' is maybe not the best here… experiments would be more appropriate.

We agree with the reviewer’s comment and have edited the manuscript accordingly.

- Figure 1A: Please indicate that PCR was used here.

We agree with the reviewer’s comment and have edited the manuscript accordingly.

- Figure1C: Y-axis is not correctly labeled, not body weight is shown.

We agree with the reviewer’s comment and have edited the figure accordingly. It is body weight and the unit was changed to (g).

- Figure 1E: Supplement would be more appropriate for this information.

We agree with the reviewer’s comment. However, we will discuss with the Editor that eventuality.

- In general, the figures are very large and could easily be reduced in size to save space.

We agree with the reviewer’s comment and have edited the manuscript accordingly.

- The legends contain far too much information, moreover the findings are described in detail and even papers are cited... This does not belong here and must be revised! The legend must show what was done, nothing more. Findings should be described in the text of the results.

- Figure 2: It is a bit confusing because the abbreviation for the proteins is used here and in the legend also the written version.

We agree with the reviewer’s comment and have edited the manuscript accordingly.

- Line 332: One ‘in’ should be deleted.

We agree with the reviewer’s comment and have edited the manuscript accordingly.

- Figure 3: Presumably after converting the graphs to PDF, display problems occurred.

We agree with the reviewer’s comment and have edited the manuscript accordingly.

- Figure 4C: The lines are difficult to distinguish... perhaps additional dashed or dotted lines could be used here.

We agree with the reviewer’s comment and have edited the manuscript accordingly.

- It is often not clear in the results how this was investigated. Isolated mitochondria from which tissue exactly? Infarct area?

We agree with the reviewer’s comment. In fact, since a global ischemia was performed, the entire heart is considered as at risk. Therefore, mitochondria were isolated from the total heart tissue.

- Figure 5: The order of the columns and the labeling is different, which is a bit confusing...

We agree with the reviewer’s comment and have edited the manuscript accordingly.

- Line 448: Fig. 8B?

We agree with the reviewer’s comment and have edited the manuscript accordingly. It is Fig. 5B.

- The graphic abstract in Figure 10 seems a bit unkind... you could do better!

We agree with the reviewer’s comment and have edited the manuscript accordingly.

Reviewer 3 Report

Dr. Feng et al. generated Mitofilin knockout mice and found that cardiac injury was increased in the knockout mice compared to wild type. They further found that the increased injury was due to sensitized MPTP opening and more ROS generation in the knockout mice. Mitochondrial damage was also increased in the knockout mice following ischemia-reperfusion compared to wild type. Dysregulation of SLC25As may be responsible for increased ROS production and MPTP opening following ischemia-reperfusion.

It is a well-written review manuscript. The reviewer has some concerns about the manuscript.

Line 156-157 “Sham hearts were not subjected to I/R but perfused for 3 hrs.” Please clarify if all sham mice underwent 3 hours perfusion in that mouse heart used for mitochondrial isolation was only reperfused for 30 minutes. Total perfusion time should be 95 (30+35+30) minutes.

2.9. Ca2+-induced mitochondrial permeability transition pore (mPTP) opening

What substrate are used in the CRC assay?

Immunoblotting image

Molecular weight should be shown on the images.

Figure 4 A and B

Could authors put arrowhead in the image to show the damaged cristate?

3.6. Mitochondria from Mitofilin+/- mice are more uncoupled and produce more reactive oxygen 404 species (ROS) following I/R.

Here authors only measured mitochondrial inner membrane potential. There were no oxidative phosphorylation and enzyme activity measurement here. Authors should do more mitochondria functional analysis to identify the defect sites in the electron transport chain. The results will help to explain why knockout of Mitofilin leads to increased ROS generation?

Please specify the substrates used for ROS measurement.

Discussion

In previous study, authors showed that the content of mitofilin was decreased in hearts following ischemia-reperfusion. Could authors discuss the potential mechanism by which ischemia-reperfusion leads to decreased mitofilin content.

Author Response

Reviewer 3

We would like to thank the reviewer for all his constructive comments.

Dr. Feng et al. generated Mitofilin knockout mice and found that cardiac injury was increased in the knockout mice compared to wild type. They further found that the increased injury was due to sensitized MPTP opening and more ROS generation in the knockout mice. Mitochondrial damage was also increased in the knockout mice following ischemia-reperfusion compared to wild type. Dysregulation of SLC25As may be responsible for increased ROS production and MPTP opening following ischemia-reperfusion.

It is a well-written review manuscript. The reviewer has some concerns about the manuscript.

Line 156-157 “Sham hearts were not subjected to I/R but perfused for 3 hrs.” Please clarify if all sham mice underwent 3 hours perfusion in that mouse heart used for mitochondrial isolation was only perfused for 30 minutes. Total perfusion time should be 95 (30+35+30) minutes.

We agree with the reviewer’s comment and have edited the manuscript accordingly. In fact, sham hearts used in the study for myocardial infarct size and cardiac function were perfused for 3 hrs without the ischemic insult. However, sham hearts used in mitochondria assessment study received 95 min of perfusion without the ischemic insult.

2.9. Ca2+-induced mitochondrial permeability transition pore (mPTP) opening

What substrate are used in the CRC assay?

As stated in the materials and methods (2.7. Mitochondrial isolation section), the buffer B used for CRC assay contains succinic acid 5 mM indicating that the substrate is succinate.

Immunoblotting image

Molecular weight should be shown on the images.

We agree with the reviewer’s comment and have edited the manuscript accordingly.

Figure 4 A and B

Could authors put arrowhead in the image to show the damaged cristate?

We agree with the reviewer’s comment and have edited the figures accordingly.

3.6. Mitochondria from Mitofilin+/- mice are more uncoupled and produce more reactive oxygen 404 species (ROS) following I/R.

Here authors only measured mitochondrial inner membrane potential. There were no oxidative phosphorylation and enzyme activity measurement here. Authors should do more mitochondria functional analysis to identify the defect sites in the electron transport chain. The results will help to explain why knockout of Mitofilin leads to increased ROS generation?

We agree with the reviewer’s comment. However, the measure of the mitochondrial inner membrane potential is a good indication of the uncoupling of mitochondria. For example when the CCCP is used in to uncouple mitochondria, there is no need to identify which complex is impaired. We will perform additional to determine which complex of the ETP is impaired in future studies. The main reason is that we consider that investigating the which complex is impaired will not add much to the overall idea of mitochondria from Mitofilin heterozygote hearts being more uncoupled when their littermate WT after I/R.

Please specify the substrates used for ROS measurement.

We mistakenly forgot to add the materials and method of the ROS measurement in the manuscript. We have added a section (2.11) describing the method used for ROS measurement that indicate the use of the substrate of complex 1, glutamate/malate (3 mM, as shown in the Fig 5B).

Discussion

In previous study, authors showed that the content of mitofilin was decreased in hearts following ischemia-reperfusion. Could authors discuss the potential mechanism by which ischemia-reperfusion leads to decreased mitofilin content.

We agree with the reviewer’s comment and have added a paragraph in the discussion.

Round 2

Author Response

  1. Revision 2 Authors speculate on an unexpected or unclear result. There are specific DRP1 post-translational modifications that determine the translocation into mitochondria, the study of which with specific antibodies can raise light on the unexpected result. Moreover, the ratio of cytosolic versus mitochondrial DRP1 can help to obtain a clear picture of the DRP1 localization. However, authors have not even tried to perform an experiment to address this specific and controversial result. Therefore my concern remains unresolved.

We do not agree with the reviewer’s comment. With all the respect to the reviewer, we don’t understand what adding cytosolic versus mitochondria DRP1 would provide to the central hypothesis of the manuscript. If the reviewer’s concern regards the mitochondrial isolation protocol, we humbly would like to refer to our twenty years’ experience in publishing in the mitochondria field and the thousand of papers published with that mitochondria isolation protocol. However, we have performed Western blot analysis just for the reviewer record to confirm that DRP1 is reduced in mitochondria from HET when compared to WT mitochondria. In addition, GAPDH present in both cytosolic fractions was almost absent in the mitochondrial fractions suggesting an absence of cytosolic contamination in the mitochondrial fraction.

Another point was how a slight decrease in fusion ad fission proteins can induce alterations. Authors still claim in the manuscript (line 312 and 326) that Mitofilin+/- heart Mitochondria exhibit impaired mitochondrial dynamics in normal 312 conditions. As commented in my previous revision, this is not supported by the data since no experiments demonstrate it ( see my first revision). Authors can only say that there is a slight decrease in MFN2, OPA AND DRP1 in the mitochondrial fraction ( in case the latter result is confirmed by proper experiments, see my previous point). Therefore, also in this case, my concern has not been addressed.

We have removed that conclusion as per the reviewer comment and edited the manuscript accordingly.

About the request of using a cytosolic marker to understand a possible loss of mitochondrial mass, obviously, have to be done in total lysates, not in the mitochondrial fraction. Therefore measuring GAPDH in isolated mitochondria is not useful. Once again, this point has not been addressed by the authors.

We present here the graph showing no difference between WT and Mic60+/- mtDNA measured by quantifying the levels of (D-LOOP1). Note that the same trend was observed with mtND4 (data not shown). This result indicates that mitochondrial mass is not changed in both sham hearts. Because two other reviewers have already validated the manuscript, we did not consider to add this data to the manuscript.

  1. Revision 2 My request was about other proteins of the Inner mitochondrial membrane (IMM) since are not directly implicated in interactions with other organelles, giving more accurate quantification of the mitochondria mass. This must be calculated by measuring the ratio with cytosolic proteins to exclude loss of mitochondrial mass. This request was also not considered by the authors, and therefore my concerns about misleading interpretation remain.

We do not agree with the reviewer’s characterization of our data. Most of the IMM proteins we tested including Cyclophilin D, VDAC1, 2, 3 or ATPase are strictly mitochondrial proteins. As shown above the levels of total mtDNA is similar in both groups, this indicate that the mitochondrial mass is similar in both group. Also since IMM protein levels are similar in both groups confirming that the mitochondrial mass is also equal in both groups. The technique of mitochondria isolation that we used exists since more than three decades and it has been used abundantly. It is scientifically not conceivable that the two groups of mitochondria could have the same IMM protein levels and have different mitochondria mass.

  1. Revision 2 Authors write in the manuscript that cristae morphology in Mitofilin+/- mice 379 was slightly disorganized, but not disrupted when compared to WT (4±0.8% versus 3±0.4%). Please, this is not slight disorganization of cristae, cristae and mitochondria appear perfectly fine in untreated samples of HET mice. What is evident, as I mentioned in the first revision, is the strong and significant alteration of the tissue structure.

We have edited that section accordingly.

  1. Revision 2 my request was a control to exclude that differences were due to alterations due to mitochondria isolation. If the differences are due to mechanical disruption during isolation or the experimental procedure, CsA should not protect. As a reviewer, I consider this kind of control essential to support data and avoid misleading interpretations.

But, again, my concerns remain. I hope the authors understand my points better this time

We understand the willing of the reviewer to add more data (controls) in our manuscript. However, we consider that CsA data included in the CRC section is enough to address the reviewer’s concern. We think that adding CsA data to measure the MMP will not change the scoop of the current manuscript. The best way to test whether the difference in the mitochondria quality is due to mechanical disruption during isolation is to measure the levels of IMM proteins in the cytosol. We have never could detect IMM proteins in the cytosol. Howerver for the reviewer’s record we provide evidence that CSA treatment preserved MMP after I/R.
